# SELECTIVE VISUAL REPRESENTATIONS IMPROVE CONVERGENCE AND GENERALIZATION FOR EMBODIED AI

**Ainaz Eftekhar**[* 1,2]    **Kuo-Hao Zeng** [* 2]    **Jiafei Duan**[1,2]
**Ali Farhadi**[1,2]    **Ani Kembhavi**[1,2]    **Ranjay Krishna**[1,2]

[1]University of Washington, [2]Allen Institute for Artificial Intelligence
{ainaze, khzeng, jiafeid, alif, anik, ranjayk}@allenai.org

https://embodied-codebook.github.io

## ABSTRACT

Embodied AI models often employ off the shelf vision backbones like CLIP to encode their visual observations. Although such general purpose representations encode rich syntactic and semantic information about the scene, much of this information is often irrelevant to the specific task at hand. This introduces noise within the learning process and distracts the agent's focus from task-relevant visual cues. Inspired by selective attention in humans—the process through which people filter their perception based on their experiences, knowledge, and the task at hand—we introduce a parameter-efficient approach to filter visual stimuli for embodied AI. Our approach induces a task-conditioned bottleneck using a small learnable codebook module. This codebook is trained jointly to optimize task reward and acts as a task-conditioned selective filter over the visual observation. Our experiments showcase state-of-the-art performance for object goal navigation and object displacement across 5 benchmarks, ProcTHOR, ArchitecTHOR, RoboTHOR, AI2-iTHOR, and ManipulaTHOR. The filtered representations produced by the codebook are also able generalize better and converge faster when adapted to other simulation environments such as Habitat. Our qualitative analyses show that agents explore their environments more effectively and their representations retain task-relevant information like target object recognition while ignoring superfluous information about other objects. Code is available on the project page.

## 1 INTRODUCTION

Human visual perception is far from a passive reception of all available visual stimuli; it is an actively tuned mechanism that operates *selectively*, allocating attention and processing stimuli that are deemed relevant to the current task (Bugelski & Alampay, 1961). An illustrative example of this phenomenon is the common experience of misplacing one's keys; we become subsequently oblivious to most visual cues in our environment, except for those directly related to the search for the lost keys. In this case, we become particularly attentive to surfaces where we usually place our keys and navigate our environment by similarly processing the walkable areas around us (Figure 1).

The subfield of embodied artificial intelligence (AI) studies AI agents tasked with very analogous situations (Duan et al., 2022). Embodied AI tasks such as navigation (Batra et al., 2020b; Krantz et al., 2023), instruction following (Anderson et al., 2018; Krantz et al., 2020; Shridhar et al., 2020), manipulation (Ehsani et al., 2021; Xiang et al., 2020), and rearrangement (Batra et al., 2020a; Weihs et al., 2021a), necessitate goal-directed behaviors, such as navigating to a specific goal or relocating target objects. Conventional frameworks usually employ general-purpose visual backbones (Khandelwal et al., 2022; Yadav et al., 2023b) to extract representations from visual input. These representations are then fused with goal embeddings (*e.g.*, object type, images, or natural language instructions) to construct a goal-conditioned representation $E \in \mathcal{R}^D$, where $D$ often has dimensions as large as a 1568-dimensional Hilbert space. $E$ captures an abundance of details from the visual input, of which a policy determines which action to take next. Given $E$'s general-purpose nature, it often contains a significant amount of task-irrelevant information. For example to find a specific object in a house, the agent doesn't need to know about other distractor objects in the agent's

---

[*]Equal contribution

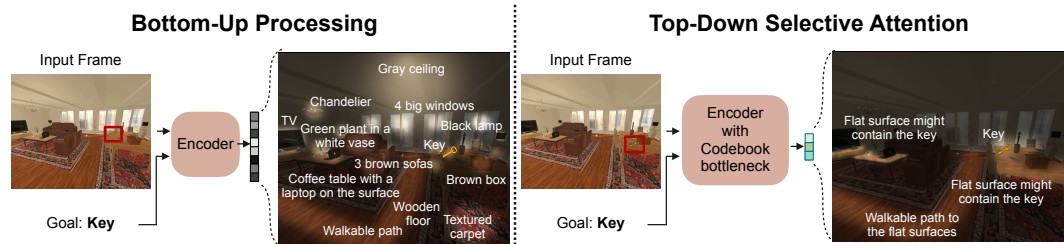

Figure 1: **Selective Attention**. Imagine an agent is tasked to locate a key in an environment. Standard visual encoders such as CLIP encoder capture general purpose scene information which include details not relevant to the task, such as the color of the sofa or texture of the floor. This mirrors the concept of bottom-up processing, where perception is influenced by external stimuli in the environment. To address this, we equip the encoder with a codebook bottleneck that only retains the most task-relevant information such as identifying flat surfaces likely to hold the key and the walkable paths to these surfaces. This represents top-down selective processing where the perception is guided by internal goals and expectations.

view, about their colors, materials, attributes, etc. These distractions introduce unnecessary noise into the learning process, distracting the agent's focus away from more pertinent visual cues.

In this paper, we draw from the substantial body of research in cognitive psychology and neuroscience to induce selective task-specific representations that filter irrelevant sensory input and only retain the necessary stimuli. In Psychology, *inattentional blindness* posits that people become "blind" to objects and details irrelevant to the task at hand (as depicted by the famous *invisible gorilla* video on YouTube[1]) (Simons & Chabris, 1999). Similar studies in *human visual search* (Wolfe, 1994; Treisman & Gelade, 1980) find that people become oblivious to the distracting items when searching through a cluttered environment. The principle of *top-down selective attention* (Desimone & Duncan, 1995; Buschman & Miller, 2007) also projects a similar theory: that our goals *bottleneck* our visual processing guided by internal goals (in contrast to bottom-up processing, which is how vision encoders are usually trained).

To apply this principle of selective attention to embodied AI agents, we propose a parameter-efficient approach to bottleneck visual representations conditioned on the task. We insert a simple *codebook module* into our agent, which consists of a collection of $K = 256$ learnable latent codes, each with a dimension of $D_c = 10$, where $D_c \ll D$. This codebook module accepts $E$ as input, attends over the $K$ latent codes, to produce a codebook representation $\hat{E}$. $\hat{E}$ is a weighted convex combination of the $K$ latent codes, weighted by the attention estimates (see Figure 2). Given the bottleneck induced by needing to choose amongst $K = 256$ possible $D_c = 10$-dimensional representations, the codebook module enforces selective filtering, encoding only the essential cues necessary for the task.

We demonstrate the effectiveness of our approach by achieving zero-shot, state-of-the-art performances on 2 Embodied-AI tasks: object goal navigation(ObjNav) (Deitke et al., 2020) and object displacement (ObjDis) (Ehsani et al., 2022) across 5 benchmarks (ProcTHOR(Deitke et al., 2022b), ArchitecTHOR, RoboTHOR (Deitke et al., 2020), AI2-iTHOR(Kolve et al., 2017a), and ManipulaTHOR (Ehsani et al., 2021)). Across all benchmarks, our approach yields significant absolute improvements in success rate and reductions in episode length. Moreover, we further show the adaptability of our codebook representations to new visual domains by lightweight finetuning in Habitat environments (Savva et al., 2019). Finally, we conduct a comprehensive analysis, verifying that the codebook encodes better information about the task, about the distance to the goal for navigation, etc. while its ability to identify individual objects in its view diminishes significantly. Surprisingly, we observe noticeable improvements in the agent's behavior in the form of smoother trajectories and more efficient exploration strategies.

## 2 RELATED WORK

We situate our work amongst methods that learn performant representations for Embodied AI tasks.
**Learning representation through proxy objectives.** A common approach to learning useful representations is defining proxy objectives for the visual encoder. For example, a paradigm outlines a set of invariant transformations and learns representations with a contrastive learning objective while simultaneously optimizing task reward (Du et al., 2021; Singh et al., 2023; Guo et al., 2018; Laskin

---

[1]Link to invisible gorilla video depicting selective attention

et al., 2020; Majumdar et al., 2022). Others use auxiliary tasks, such as predicting depth (Mirowski et al., 2017; Gordon et al., 2019), reward (Jaderberg et al., 2017), forward dynamics (Gregor et al., 2019; Zeng et al., 2023; 2021; Guo et al., 2020; Kotar et al., 2023), inverse dynamics (Pathak et al., 2017; Ye et al., 2021b;a), scene graphs (Gadre et al., 2022; Du et al., 2020; Savinov et al., 2018; Wu et al., 2019), or 2D maps (Chaplot et al., 2020b;a). In contrast, our proposed codebook module is jointly trained with the overall policy to optimize the task rewards only.

**Learning representations through pretraining.** Alternately, many proposed methods self-supervise representations by sampling image frames from the environment during a pre-training stage (Yadav et al., 2023b;a; Kotar et al., 2023; Mezghan et al., 2022). Some utilize 3D constraints to induce 3D-awareness into the visual representations (Wallingford et al., 2023; Marza et al., 2023). There are even been complicated games designed to encourage the emergence of useful representations (Weihs et al., 2021b). Of particular interest is the EmbCLIP (Khandelwal et al., 2022) model, which serves as our main baseline. This method leverages the large-scale, pretrained representations by CLIP (Radford et al., 2021) as the visual encoder. Our model builds upon EmbCLIP, incorporating the proposed codebook module to filter out redundant stimuli.

**Bottleneck and regularize representation.** Bottlenecked representations are common in machine learning (Hinton et al., 2011; Kingma & Welling, 2014; Alemi et al., 2017; Koh et al., 2020; Federici et al., 2020; Mairal et al., 2008), computer vision (He et al., 2016; Srinivas et al., 2021; Riquelme et al., 2021; Van Den Oord et al., 2017; Chen et al., 2023; Kolesnikov et al., 2022; Hsu et al., 2023), natural language processing (Shazeer et al., 2017; Fedus et al., 2022; Mahabadi et al., 2021), and even reinforcement learning (Serban et al., 2020; Fan & Li, 2022; Pacelli & Majumdar, 2020). The current most popular bottlenecked representation is VQ-GAN (Esser et al., 2021). Nonetheless, relatively few have implemented these techniques in Embodied AI. Most applications of information bottlenecks in this space, as far as we are aware, have been applied to full-observable MiniGrid (Chevalier-Boisvert et al., 2023) or Atari (Bellemare et al., 2013) environments (Goyal et al., 2019; Igl et al., 2019; Bai et al., 2021; Lu et al., 2020). By contrast, we situate our work within the partially-observed, physics-enabled, and visually rich Embodied AI environments. The closest related work is a recent proposal for learning a codebook for embodied AI using a NERF-based rendering objective (Wallingford et al., 2023); this method pre-trains the codebook and later uses it for downstream tasks. By contrast, we learn our codebook representations while simultaneously learning the policy to act.

## 3 METHODOLOGY

We first provide a brief background on how embodied AI agents are designed today. Next, we introduce our contributions to make the model. We lay out the details of our task-conditioned codebook and also describe our training strategy.

**Background.** The modern design for embodied agents share a common core architecture (Batra et al., 2020b; Khandelwal et al., 2022). This architecture contains three main components. First, three encoders encode the necessary information required to take an action. A visual encoder transforms visual observations (*e.g.*, RGB or depth) into a visual representation $v$; a goal encoder transforms the task's objective (*e.g.*, GPS location, target object type, or natural language instructions) into a goal embedding $g$; a previous-action encoder to embed the most recently executed action into an action embedding $\alpha$. For the visual encoder $v$, we mainly use the RGB sensors as visual stimuli. The only exception is the object displacement task, for which we follow prior work to also include a depth sensor, the segmentation mask sensor, and the end-effector sensor (Ehsani et al., 2022). The second component of the architecture fuses $v, g, \alpha$ together. The three representations are flattened and then concatenated to form the *task-conditioned representation E*. Finally, the third component keeps track of the history of the agent's trajectory and proposes the next action. This component includes a recurrent state encoder to encode and remember past steps. So, $r = E_t | t = 1, ..., T$ encodes all the past representations into a single state representation. This is injected by an actor-critic head, which predicts a distribution over action space and also produces a value estimation of the current state.

**The need for task-bottlenecked visual representations.** EmbCLIP (Khandelwal et al., 2022) is today's state-of-the-art model for the tasks that we consider which encodes the three representations in $E \in \mathcal{R}^D$, where $D = 1568$. This embedding contains CLIP features, which were trained for general-purpose vision tasks and guided through language self-supervision (Radford et al., 2021). Therefore, when presented with the appropriate input image, this representation can identify a large number of object categories, their attributes, their spatial relationships, etc. It can even encode the materials and textures of objects, and outline the actions that people appear to be taking in the image. However, for tasks in Embodied AI, where the agent is asked to locate a pair of, say, lost "keys",

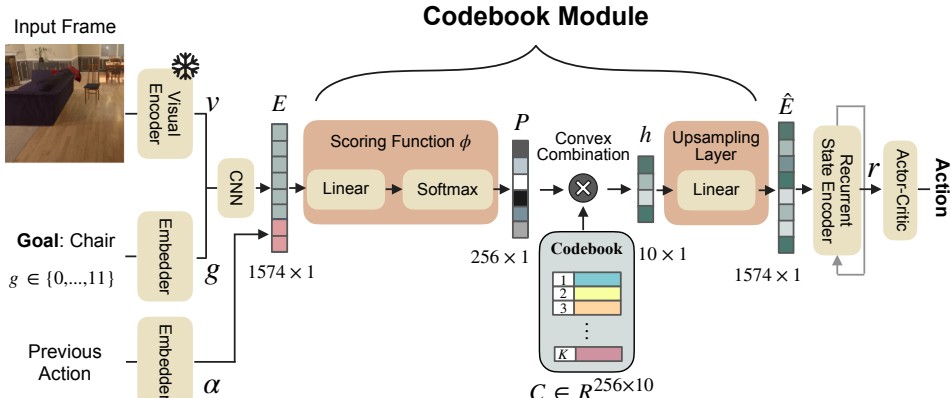

Figure 2: **An overview of EmbCLIP-Codebook.** The 3 representations corresponding to the input frame, the goal, and the previous action get concatenated to form $E \in \mathcal{R}^{1568}$. The codebook module takes $E$ and generates a probability simplex $\mathcal{P} \in \mathcal{R}^{256}$ over the latent codes. The hidden compact representation $h \in \mathcal{R}^{10}$ is a convex combination of the codes weighted by $\mathcal{P}$. The final task-bottlenecked codebook representation $\hat{E}$ is derived by upsampling $h$ which is subsequently passed to the recurrent state encoder and the policy to produce an action.

knowing the texture of the couch in the living room or the color of the flower vase on the dining table seems irrelevant. These additional pieces of information, therefore, are also sent over to the policy, which is often designed as a recurrent neural network (RNN) followed by a small actor-critic module. These few parameters, therefore, must serve two purposes: they must identify what information is useful for the task at hand and what action to take given that information.

**The codebook module.** We introduce a module that decouples the two objectives across different parameters in the network. The input encoders and the codebook focus on extracting essential information for the task from the visual input, whereas the policy (RNN and actor-critic heads) can focus on taking actions conditioned on this filtered information.

We design the codebook to be a parameter-efficient module to transform the general-purpose representation $E$ into a compact task-bottlenecked one. The module inputs $E$ and generates a compact $\hat{E}$ (see Figure 2). This module contains codes, defined as a set of latent vectors $C = [c_1, c_2, ..., c_K] \in \mathcal{R}^{K \times D_c}$ where $K$ denotes codebook's size and $D_c$ is the dimensionality of each latent code $c_k$. To create a strong bottleneck, we set $D_c = 10$ and $K = 256$. These codes are initialized randomly via a normal distribution and optimized along with the overall training algorithm.

To extract $\hat{E}$ from $E$, the module contains a scoring function $\phi(.)$ and an upsampling layer. We first use $\mathcal{P} = \phi(E)$ to generate a probability simplex over the $K$ latent codes, where $\mathcal{P} = [p_i]_{i=1}^K$ such that $\sum_{i=1}^K p_i = 1$. The scoring function $\phi$ is implemented as a single-layer MLP followed by a softmax function. This configuration forces the agent to select which latent code(s) are more useful for representing the current frame. Next, the hidden compact representation $h$ is derived by taking a convex combination of the learnable codes $\{c_i\}_{i=1}^K$ weighted by their corresponding $p_i$: $h = \mathcal{P}^T C = \sum_{i=1}^K p_i.c_i$. Further, the upsampling layer, implemented as a linear layer, upsamples the hidden embedding $h$ to the task-bottlenecked codebook representation $\hat{E}$.

By design, we place the codebook module immediately before the RNN and after the $E$ is fused. In this way, the module focuses on single-step processing, prompting the codebook to compile information at each individual step rather than relying on past steps. By positioning the codebook module right after $E$, it fuses the visual representation $v$ with the goal $g$. We compare our design choice with the codebook module positioned right after the visual encoder (without goal-conditioning) in Sec. A.9. In essence, this module design is reminiscent of memory networks, which keep track of information (Han et al., 2020), except without the write functionality; not requiring a write operation makes sense because we make the assumption that everything the agent needs to act is already encoded in the CLIP embeddings.

**Training algorithm.** We employ Proximal Policy Optimization (PPO) (Schulman et al., 2017)[2] to train the agent. PPO is an on-policy reinforcement learning algorithm, which optimizes all three components of the architecture, including the codebook module. One critical challenge we

---

[2]We use DD-PPO to optimize the policy for the object displacement task (Ehsani et al., 2022).

Table 1: Our method with the codebook outperforms the baselines on 2 different tasks, including zero-shot evaluation on 4 Object Goal Navigation benchmarks and results on Object Displacement benchmark. We use ↑ and ↓ to denote if larger or smaller values are preferred.

| Benchmark | Model | Object navigation | | | | |
| | | SR(%)↑ | EL↓ | Curvature↓ | SPL↑ | SEL↑ |
| --- | --- | --- | --- | --- | --- | --- |
| ProcTHOR-10k (validation) | EmbCLIP | 67.70 | 182.00 | 0.58 | **49.00** | 36.00 |
| | +codebook | **73.72** | **136.00** | **0.23** | 48.37 | **43.69** |
| ARCHITECTHOR (0-shot) | EmbCLIP | 55.80 | 222.00 | 0.49 | **38.30** | 20.57 |
| | +Codebook | **58.33** | **174.00** | **0.20** | 35.57 | **28.31** |
| RoboTHOR (0-shot) | EmbCLIP | 51.32 | - | - | **24.24** | - |
| | +Codebook | **55.00** | - | - | 23.65 | - |
| AI2-iTHOR (0-shot) | EmbCLIP | 70.00 | 121.00 | 0.29 | **57.10** | 21.45 |
| | +Codebook | **78.40** | **86.00** | **0.16** | 54.39 | **26.76** |

| | | Object displacement | |
| | | PU(%)↑ | SR(%)↑ |
| --- | --- | --- | --- |
| ManipulaTHOR | m-VOLE | 81.20 | 59.60 |
| | +Codebook | **86.00** | **65.10** |

encounter when training codebooks is *codebook collapse* (Van Den Oord et al., 2017; Riquelme et al., 2021; Shazeer et al., 2017), where only a handful of codes are used by the agent, limiting the model's full capacity. While several solutions[3] have been proposed to this problem, we find that using dropout (Srivastava et al., 2014) is the straightforward yet effective approach to overcome this challenge. Instead of directly employing the probability scores $\mathcal{P}$ for the convex combination, we first apply dropout to it with a rate of $0.1$. This produces new probability scores $\hat{\mathcal{P}}$, which we then use to combine the latent codes: $h = \hat{\mathcal{P}}^T C$. Intuitively, this equates to randomly setting $10\%$ of the latent codes to zero before integrating them through a convex combination, preventing the agent from relying on only a handful of codes.

## 4   EXPERIMENTS AND ANALYSIS

In our experiments, we demonstrate that the goal-conditioned codebook embedding $\hat{E}$ results in significant improvements over the non-bottlenecked embedding $E$ across a variety of Embodied-AI benchmarks (Sec. 4.1). We further show that our bottlenecked embeddings are more generalizable and can be applied across different domains without exhaustive finetuning (Sec. 4.2). We study (both qualitatively and quantitatively) the visual cues encoded in the codebook and how they relate to the target task (Sec. 4.3). Finally, we evaluate whether our approach is suitable for other pretrained visual encoders. We substitute the CLIP representations with DINOv2 (Oquab et al., 2023) visual features and replicate our experiments. The results confirm that our proposed codebook module is representation-agnostic and can effectively bottleneck various pretrained visual representations.

### 4.1   CODEBOOK-BASED REPRESENTATIONS IMPROVE PERFORMANCE IN EMBODIED-AI

**Tasks.** We present our results on several navigation benchmarks and a manipulation benchmark to demonstrate the benefits of compressing the task-conditioned embedding using the codebook. We consider ObjectNav (navigate to find a specific object category in a scene) in PROCTHOR (Deitke et al., 2022a), ARCHITECTHOR, ROBOTHOR (Deitke et al., 2020), and AI2-iTHOR (Kolve et al., 2017b). We also consider Object Displacement (bringing a source object to a destination object using a robotic arm) (Ehsani et al., 2022) in ManipulaTHOR (Ehsani et al., 2021) as our manipulation task.

#### 4.1.1   OBJECT NAVIGATION

**Model.** We adopt the same core architecture used in EmbCLIP (Khandelwal et al., 2022) for Object Navigation. RGB images are encoded using a frozen CLIP ResNet-50 model to a $2048 \times 7 \times 7$ tensor and then compressed by a 2-layer CNN to another $32 \times 7 \times 7$ tensor $v$. This tensor is concatenated with the 32-dim goal embedding $t$ and a previous action embedding $\alpha$, passed through another 2-layer CNN and flattened to form a $D = 1568$-dim goal-conditioned observation embedding $E$. While EmbCLIP directly passes this 1568-dim representation, $E$, to a 1-layer recurrent state encoder with 512 hidden units, we use a codebook module with codebook's size of $K = 256$ and $D_c = 10$ as a bottleneck to compress this goal-conditioned embedding and filter out the irrelevant information. We further pass the resulting goal-bottlenecked codebook embedding $\hat{E}$ to the following recurrent state encoder. We term our model as *EmbCLIP-Codebook* in the rest of the paper.

---

[3]We tried Linde-Buzo-Gray (Linde et al., 1980) splitting algorithm and Gumbel Softmax (Jang et al., 2017) with careful temperature tunning, but have found that the dropout is the most effective approach.

Table 2: Results for models trained on ProcTHOR and evaluated with fine-tuning on Habitat Object Goal Navigation benchmarks. Our method with the codebook can adapt to Habitat environment through the lightweight finetuning. We use ↑ and ↓ to denote if larger or smaller values are preferred.

| Benchmark | Fine-tuning parts | Model | Object goal navigation | | | |
|---|---|---|---|---|---|---|
| | | | SR(%)↑ | SPL↑ | Invalid Actions(%)↓ | Curvature↓ |
| Habitat challenge 2022 (HM3D Semantics) | Adaptation Module | EmbCLIP | 36.45 | 18.18 | 28.10 | 0.53 |
| | | +Codebook | **50.25** | **25.76** | **21.50** | **0.26** |
| | Entire Model | EmbCLIP | 58.00 | 30.97 | 15.80 | 0.52 |
| | | +Codebook | 55.00 | 29.21 | 15.40 | 0.38 |

**Experiment Details**. All models are trained using the AllenAct framework. We follow (Deitke et al., 2022b) to pretrain the Emb-CLIP baseline and EmbCLIP-Codebook on the the PROCTHOR-10k houses with 96 samplers for 435M steps. The models are evaluated zero-shot on the downstream benchmarks, including PROCTHOR-10k val scenes, ARCHITECTHOR val scenes, ROBOTHOR test scenes[4], and AI2-iTHOR val scenes. Each model is evaluated for 5 checkpoints between 415M to 435M training steps and we select the best model results from the ARCHITECTHOR val scenes. Please refer to the supplementary Sec. A.1 for more training details.

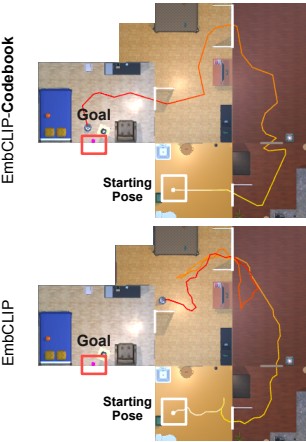

**Result.** Table 1 shows EmbCLIP-Codebook achieves best *Success Rate (SR)* across all 4 benchmarks. Furthermore, *Episode Length (EL)* is significantly lower compared to EmbCLIP suggesting that our agent is able to find the goal in much fewer steps. We further present *Curvature* as a key evaluation metric comparing the smoothness of the trajectories. Smoother trajectories are generally safer and more energy-efficient. Excessive rotations and sudden changes in direction can lead to increased energy consumption and increase the chances of collisions with other objects. The curvature value $k$ is defined as $k = \frac{dx \times ddy - dy \times ddx}{(dx^2 + dy^2)^{\frac{3}{2}}}$ for each coordinate $(x, y)$ on the trajectory and

Figure 3: **Sample Trajectory.** EmbCLIP agent takes many redundant rotations, resulting in a high average curvature, whereas ours navigates more smoothly.

it's averaged across all time steps in the agent's path. A higher value of average *Curvature* signifies more rotations and changes in direction. Conversely, a value closer to zero suggests a predominantly straight and smooth trajectory. Our agent travels in much smoother paths across all benchmarks. Figure 3 shows an illustrative example. Please find more examples in Supplementary Sec. A.7.

**SPL Limitations and Introducing a New Metric SEL.** We further report the *Success Weighted by Path Length (SPL)* as one of the common metrics used in the field for Object Navigation task. *SPL* is defined as $\frac{1}{N} \sum_{i=1}^{N} S_i \frac{l_i}{max(l_i, p_i)}$ where $l_i$ is the shortest possible path (traveled distance) to the target object, $p_i$ is the taken path by the agent and $S_i$ is a binary indicator of success for episode $i$. As shown in the table, EmbCLIP-Codebook outperforms in Success Rate and solves the task with fewer steps (smaller EL). However, it lags behind in the SPL metric. This discrepancy arises because SPL is evaluated based on the distance traveled rather than the actual number of steps taken. This highlights a limitation in this metric since the efficiency of a path should consider factors like time and energy consumption. Actions such as rotations and look-ups/downs, which also consume time and energy, are not accounted for in this metric. In light of this observation, we further report *Success Weighted by Episode Length (SEL)*: $\frac{1}{N} \sum_{i=1}^{N} S_i \frac{w_i}{max(w_i, e_i)}$, where $w_i$ is the shortest possible episode length to the target object, and $e_i$ is the episode length produced by the agent. We utilize the privileged information from the environment to develop an expert agent to collect $w_i$ for each episode. Shown in Table 1, EmbCLIP-Codebook outperforms EmbCLIP by a significant margin in *SEL*.

### 4.1.2 OBJECT DISPLACEMENT

**Model.** We adopt the same model architecture (m-VOLE w. GT mask) used in m-VOLE (Ehsani et al., 2022). The model consists of a 3-layers CNN to encode the RGB-D and a ground truth segmentation mask, a frozen ResNet-18 to encode the query images of target classes, including the source object (i.e., an apple) as the object has not picked up yet or the destination object after the source object (i.e., a table) has been picked up, and a distance embedder to encode the relative distance computed by the Object Location Estimator Module, which measures the distance between

---

[4]Link to RoboTHOR leaderboard

the end-effector and the source/destination object. The resulting visual features of current observation and query image are then concatenated with the distance feature to construct the goal-conditioned observation embedding $E$ with $D = 1536$. While m-VOLE directly passes the embedding $E$ to the following recurrent state encoder, we add our codebook module with codebook's size of $K = 256$ and $D_c = 10$ to compress the embedding $E$ to the goal-conditioned codebook embedding $\hat{E}$. Finally, the recurrent state encoder processes the embedding $\hat{E}$ and the actor-critic after the recurrent state encoder further predicts the action probability distribution as well as the state value estimation. We call our model as *m-Vole-Codebook* in the rest of the paper.

**Experiment Details.** We, again, use the AllenAct framework to train our m-Vole-Codebook with the default reward shaping provided by (Ehsani et al., 2022). Following (Ehsani et al., 2022), we train our agent by DD-PPO (Wijmans et al., 2019) with 80 samplers for 20M steps on APND dataset (Ehsani et al., 2021) in 30 kitchen scenes, where we split them into 20 training scenes, 5 validation scenes, and 5 testing scenes. During the training stage, the agent is required to navigate to the source object, pick it up, and bring it to the destination object. Please refer to Sec. A.1 for more training details.

**Results.** To quantify the models' performance, we report *PickUp Success Rate (PU)* and *Success Rate (SR)* as our evaluation metrics. As shown in Table 1, we can find that our m-Vole-Codebook outperforms m-VOLE baseline on both *PU* and *SR*. It demonstrates that our codebook module is applicable to the Embodied AI tasks involving interactions between the agent and environment.

## 4.2 CODEBOOK EMBEDDING IS EASIER TO TRANSFER TO NEW VISUAL DOMAINS

**Model.** This section investigates the capability of the codebook embedding to transfer across new visual domains without exhaustive finetuning. Efficient transfer of Embodied-AI agents between distinct visual domains remains a important challenge for many downstream applications. For example, there might be a desire to retain the high-level decision-making policy established in the recurrent state encoder and the actor-critic during large-scale pretraining. Alternatively, resource constraints may prevent end-to-end finetuning of the entire model in the target domain. To address this, we propose finetuning only the CNN layers that follow the frozen CLIP ResNet backbone, the goal encoder, and the previous action encoder. This way ensures adjustments only to the visual representation and the task embedding, thereby adapting the agent to the target domain without changing the policy's behaviors completely. We call these fine-

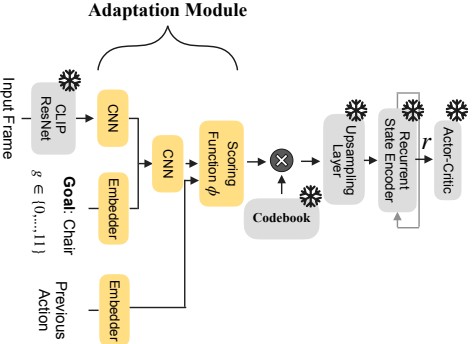

Figure 4: **Lightweight Finetuning of the Adaptation Module.** We only finetune a few CNN layers, action/goal embedders, and the codebook scoring function when moving to new visual domain.

tuned components as the *Adaptation Module*, shown in Figure 4. Meanwhile, other components, including the CLIP ResNet, codebook, recurrent state encoder, and actor-critic, remain frozen.

**Experiment Details.** We conduct lightweight finetuning of the *Adaptation Module* in both EmbCLIP-Codebook and the baseline for Object Navigation on Habitat 2022 ObjectNav challenge. Since AI2THOR and Habitat simulators exhibit variations in visual characteristics, lighting, textures and other environmental factors, they are appropriate platforms for studying the lightweight transfer capabilities of Embodied-AI agents. Following (Deitke et al., 2022b), we use the checkpoint at 200M steps to perform the finetuning in Habitat for another 200M steps with 40 samplers. Since the target object types in Habitat are subset of the ones in PROCTHOR, we discard the parameters in the goal embedder corresponding to the unavailable object types (more details in Supplementary Sec. A.1).

**Results.** Table 2 shows that EmbCLIP-Codebook achieves superior performance during the lightweight fine-tuning of the Adaptation Module across all metrics. To elaborate, we attain a 13.8% higher success rate and a 7.58% improvement in SPL, all while executing fewer invalid actions and achieving smoother trajectories. Our codebook bottleneck effectively decouples the process of learning salient visual information for the task from the process of decision-making based on this filtered information. Consequently, when faced with a similar task in a new visual domain, the need for adaptation is significantly reduced. In this scenario, only the modules responsible for extracting essential visual cues in the new domain require fine-tuning, while the decision-making modules

Figure 5: **GradCAM Attention Visualization.** While EmbCLIP is distracted by different objects and other visual cues even though the target object is visible in the frame, EmbCLIP-Codebook is able to effectively ignore such distractions and only focus on the object goal.

Table 3: Linear probing shows codebook embeddings encode more task-relevant cues

| Task | Model | Accuracy↑ | F1 Score↑ |
|------|-------|-----------|-----------|
| Object Presence | EmbCLIP | **0.10** | **0.55** |
| (all 125 categories) | +Codebook | 0.05 | 0.29 |
| Object Presence | EmbCLIP | **0.36** | **0.53** |
| (16 goal categories) | +Codebook | 0.27 | 0.11 |
| Goal Visibility | EmbCLIP | 0.87 | - |
| (binary) | +Codebook | **0.92** | - |
| Distance to Goal | EmbCLIP | 0.29 | - |
| (5-way classification) | +Codebook | **0.31** | - |

Table 4: Our method with the codebook consistently outperforms the baselines using DINOv2 (Oquab et al., 2023) visual features in zero-shot evaluation on 4 Object Goal Navigation benchmarks.

| Benchmark | Model | Object navigation | | | | |
|-----------|-------|-------|-----|-----------|------|------|
| | | SR(%)↑ | EL↓ | Curvature↓ | SPL↑ | SEL↑ |
| ProcTHOR-10k (validation) | DINOv2 (Oquab et al., 2023) | 74.25 | 151.00 | 0.24 | 49.53 | 43.20 |
| | +Codebook (Ours) | **76.31** | **129.00** | **0.12** | **50.26** | **44.70** |
| ARCHITECTHOR (0-shot) | DINOv2 | 57.25 | 218.00 | 0.25 | 36.83 | 29.00 |
| | +Codebook (Ours) | **59.75** | **194.00** | **0.11** | 36.00 | **31.70** |
| AI2-iTHOR (0-shot) | DINOv2 | 74.67 | 97.00 | 0.19 | 59.45 | 26.50 |
| | +Codebook (Ours) | **76.93** | **68.00** | **0.07** | **60.14** | **28.30** |
| RoboTHOR (0-shot) | DINOv2 | 60.54 | - | - | **29.36** | - |
| | +Codebook (Ours) | **61.03** | - | - | 28.01 | - |

can remain fixed. In contrast, EmbCLIP, which does not decouple this skill learning, necessitates adaptation of both visual cue extraction and decision-making modules when transitioning to a new domain. This is the primary reason for the substantial performance gap observed in the lightweight fine-tuning setting. These results are upper-bounded by end-to-end finetuning of the entire model with access to more training resources. Here, EmbCLIP-Codebook performs nearly on par with the EmbCLIP while still maintaining smoother trajectories, as evidenced by the curvature metric.

### 4.3 CODEBOOK ENCODES ONLY THE MOST IMPORTANT INFORMATION TO THE TASK

We conduct an analysis (both qualitatively and quantitatively) to explore the information encapsulated within our bottlenecked representations after training for Object Navigation on PROCTHOR-10k.

**Linear Probing.** We use linear probing to predict a list of primitive tasks from the goal-conditioned visual embedding $E$ in EmbCLIP and the bottlenecked version $\hat{E}$ in EmbCLIP-Codebook generated for 10k frames in ARCHITECTHOR scenes. The selected tasks include *Object Presence* (identifying which of the 125 object categories are present in the image), *Goal Presence* (identifying which of the 16 goal categories are present in the image), *Goal Visibility* (determining if the object goal is visible in the frame and within 1.0m distance from the agent), and *Distance to Goal* (predicting the distance to the target object only when it's closer than 5m to the agent). We train simple linear classifiers for each of these tasks to predict the desired outcome from the embeddings. The results are summarized in Table 3. Our goal-bottlenecked representations effectively exclude information related to object categories other than the specified goal. Consequently, although our performance in predicting the presence of all object categories may be suboptimal, this information is usually irrelevant to the object navigation task. Instead our method improves accuracy in predicting goal visibility. Furthermore, our representations better encode information about the distance to the goal (when it's close enough to the agent) resulting in better navigation towards the goal in the final stages of the task.

**Grad-Cam Visualization.** We utilize Grad-Cam (Selvaraju et al., 2017) to visualize the attention map in the visual observation. Given an action predicted by the policy model at a single step, we treat it as a classification target and calculate the gradients to minimize the classification objective. With the gradients, we further apply *XGradCAM* to visualize the final CNN layer producing goal-conditioned embeddings $E$. As shown in Figure 5, we observe that EmbCLIP often focuses on many different objects, including their appearance, texture, or other distracting visual cues, even though the target object has been visible in view. However, EmbCLIP-Codebook is able to ignore many distracting visual cues and concentrate on the target object.

**Nearest Neighbor Visualization** To visually assess the information captured in our embeddings, we employ a nearest neighbor retrieval using a sample of 10k frames from Procthor, each accompanied

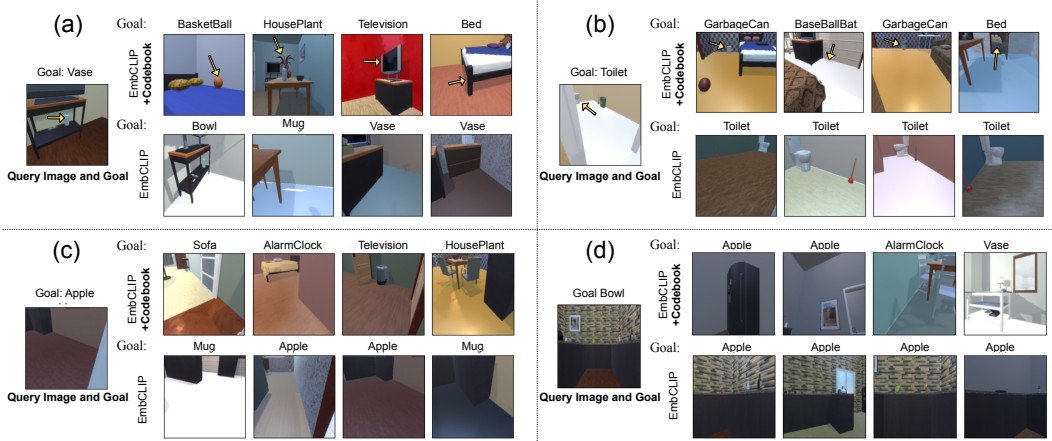

Figure 6: **Nearest-Neighbor Retrieval in the Goal-Conditioned Embedding Space ($E$ and $\hat{E}$).** The 4 examples show that EmbCLIP-Codebook prioritizes *task semantics* while EmbCLIP focusses on *scene semantics*. In (a) and (b), our nearest neighbors are based on goal visibility and goal proximity to the agent whereas EmbCLIP ones are based on the overall semantics of the scene (tables in the left, toilets far away). In (c) and (d), ours favor the overall scene layout whereas EmbCLIP mostly focuses on colors and appearances.

by a corresponding goal specification, which may or may not be visible in the frame. We feed these frames and goals into our pretrained Objectnav models, generating their respective goal-conditioned embeddings (denoted as $E$ in EmbCLIP and $\hat{E}$ in EmbCLIP-Codebook). We then identify nearest neighbors in both the $\hat{E}$ and $\hat{E}$ embedding spaces, as illustrated in Figure 6 (a) - (d).

In (a), the codebook-based nearest neighbors for the *Vase* query image are other instances of observations with the goal prominently visible, including goals that are not *Vase*. Similarly in (b), the nearest neighbors are observations with the goal visible, but now at a further distance from the agent, akin to the query image. In contrast, the EmbCLIP neighbors for (a) include a wooden table on the left, like the query image, and in (b) include free space with a toilet in the distance. These examples indicate that EmbCLIP-Codebook prioritizes *task semantics* over EmbCLIP which tends to focus on *scene semantics*. In (c) and (d), the query does not contain the goal object. Here, EmbCLIP-Codebook neighbors show a similarity in the overall scene layout, where as EmbCLIP neighbors tend to prioritize the appearance of the floor and walls.

## 4.4 CODEBOOK MODULE IS REPRESENTATION-AGNOSTIC

To assess the applicability of the codebook to other visual encoders and to quantify the extent of the improvements irrespective of the underlying representation, we replaced the CLIP representations with DINOv2 (Oquab et al., 2023) visual features. DINOv2 relies on discriminative self-supervised pre-training (which is different from the image-text pretraining approach utilized in CLIP) and is proven to be effective across a wide range of tasks, including Embodied-AI. We use the frozen DINOv2 ViT-S/14 model to encode RGB images into a $384 \times 7 \times 7$ tensor which is compressed to tensor $v$ using a 2-layer CNN. Similar to EmbCLIP, $v$ is concatenated with a 32-dimensional goal embedding and the previous action embedding and the result is flattened to obtain a 1574-dimensional goal-conditioned observation embedding, denoted as $E$. We employed a codebook with similar dimensions, $K = 256$ and $D_c = 10$, to bottleneck the goal-conditioned representations. The results are presented in Table 4. Consistently, our approach outperforms the DINOv2 baseline models across a variety of Object Navigation metrics in various benchmarks. This experiment underscores the effectiveness of our codebook module in bottlenecking other visual features for embodied-AI tasks.

## 5 CONCLUSION

Inspired by selective attention in humans, we proposed a compact learnable codebook module for Embodied-AI that decouples identifying the salient visual information useful for the task from the process of decision-making based on that filtered information. It acts as a task-conditioned bottleneck that filters out unnecessary information, allowing the agent to focus on more task-related visual cues. It significantly outperforms state-of-the-art baselines in Object goal navigation and Object displacement tasks across 5 benchmarks. This results in significantly faster adaptation to new visual domains. Our qualitative and quantitative analyses show that it captures more task-relevant information, resulting in more effective exploration strategies.

ACKNOWLEDGMENTS

We thank members from RAIVN Lab at the University of Washington and PRIOR team at Allen Institute for AI for valuable feedbacks on this project. This work is in part supported by NSF IIS 1652052, IIS 17303166, DARPA N66001-19-2-4031, DARPA W911NF15-1-0543 and gifts from Allen Institute for AI.

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

# A    APPENDIX

The following items are provided in the Appendix:

  VI. An ablation study on codebook regularization and codebook collapse (§ A.6).

## A.1    TRAINING DETAILS

Code and pretrained models are publicly available through the project page.

**Object Goal Navigation**

We follow (Deitke et al., 2022b) to pretrain the EmbCLIP baseline and EmbCLIP-Codebook on the the PROCTHOR-10k houses using the default action space {MoveAhead, RotateRight, RotateLeft, LookUp, LookDown, Done}. During this stage, we optmize the models by Adam (Kingma & Ba, 2014) optimizer with a fixed learning rate of $3e^{-4}$. In addition, we follow (Deitke et al., 2022b) to have two warm up stages by training model parameters with lower number of steps per batch for PPO training (Schulman et al., 2017). In the first stage, we set number of steps as 32, and in the second stage, we increase the number of steps to 64. These two stages are trained by 1M steps, respectively. After the second stage, we increase the number of steps to 128 and keep it till the end of training (*e.g.*, 435M steps). For reward shaping, we employ the default reward shaping defined in AlleAct: $R_{penalty} + R_{success} + R_{distance}$, where $R_{penalty} = -0.01$, $R_{success} = 10$, and $R_{distance}$ denotes the cahnge of distances from target between two consecutive steps. In the experiments, we follow (Deitke et al., 2022b) to use 16 objects as possible target objects, include {AlarmClock, Apple, BaseballBat, BasketBall, Bed, Bowl, Chair, GarbageCan, HousePlant, Laptop, Mug, Sofa, SprayBottle, Television, Toilet, Vase}. Figure 7 compares the Success Rate training curves for EmbCLIP and EmbCLIP-Codebook. As shown in the figure, adding our codebook bottleneck significantly improves the sample efficiency through faster convergence, makes the training more robust, and achieves a better performance by the end of the training.

**Object Displacement**

Following (Ehsani et al., 2022), we optimize our agent's model parameters by Adam optimizer (Kingma & Ba, 2014) with a fixed learning of $3e^{-4}$ and number of steps of 128 for DD-PPO (Wijmans et al., 2019) training. On APND dataset (Ehsani et al., 2021) we use 12 possible target objects, including {Apple, Bread, Tomato, Lettuce, Pot, Mug, Potato, Pan, Egg, Spatula, Cup, and SoapBottle. During the training stage, we sample an initial location from 130 possible agent initial locations to spawn the agent and sample initial locations for source and target objects from 400 possible object-pair locations to place objects. The action space in this environment is {MoveAhead, RotateRight, RotateLeft, MoveArmBaseUp, MoveArmBaseDown, MoveArmAhead, MoveArmBack, MoveArmRight, MoveArmLeft, MoveArmUp, MoveArmDown}.

**Habitat Finetuning**

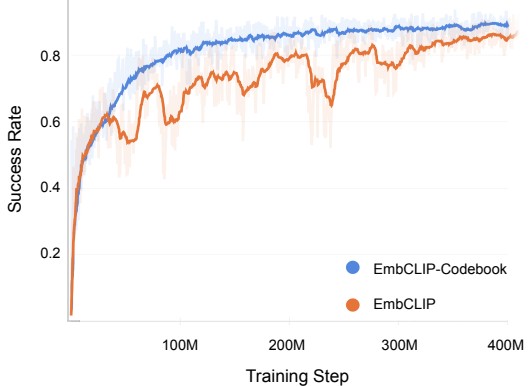

Figure 7: Success Rate training curve comparing EmbCLIP and EmbCLIP-Codebook

Table 5: Our proposed codebook module consistently outperforms other information bottleneck baselines across different ObjectNav benchmarks. EmbCLIP-AE uses an auto-encoder to bottleneck the task-conditioned embedding $E$ and EmbCLIP-SelfAttn applies self attention on top of the goal-conditioned CLIP feature maps. (Models are evaluated at 210M training steps.)

| Benchmark | Model | Object navigation | | |
|---|---|---|---|---|
| | | SR(%)↑ | EL↓ | SPL↑ |
| ProcTHOR-10k (validation) | EmbCLIP | 63.43 | 133.00 | 45.98 |
| | EmbCLIP-AE | 59.35 | 145.00 | 42.14 |
| | EmbCLIP-SelfAttn | 56.50 | **126.00** | 44.02 |
| | **EmbCLIP-Codebook** (Ours) | **72.49** | 134.00 | **49.37** |
| ARCHITECTHOR (0-shot) | EmbCLIP | 52.67 | 172.00 | 34.93 |
| | EmbCLIP-AE | 48.67 | 166.00 | 32.58 |
| | EmbCLIP-SelfAttn | 50.00 | **151.00** | **36.32** |
| | **EmbCLIP-Codebook** (Ours) | **55.25** | 160.00 | 35.31 |
| AI2-iTHOR (0-shot) | EmbCLIP | 64.13 | 108.00 | 52.76 |
| | EmbCLIP-AE | 58.93 | 96.00 | 47.29 |
| | EmbCLIP-SelfAttn | 59.07 | **77.00** | 52.30 |
| | **EmbCLIP-Codebook** (Ours) | **74.80** | 81.00 | **57.79** |

As mentioned in the main paper (Sec. 4.2), we follow (Deitke et al., 2022b) to use the checkpoint at 200M steps to perform another 200M finetuning. However, in our lightweight finetuning, we only update the model parameters inside the adaption module (shown in Fig. 4). In this finetuning stage, we use Adam (Kingma & Ba, 2014) optimizer to update the model parameters and we set the number of steps in the PPO (Schulman et al., 2017) training as 128. We try 4 learning rates, including $\{1e^{-4}, 2e^{-4}, 3e^{-4}, 4e^{-4}\}$ and find $1e^{-4}$ works best for EmbCLIP and $3e^{-4}$ works best for the model with our codebook module. Furthermore, as mentioned in the Sec. 4.2, because the target object types in Habitat, including {Bed, Chair, HousePlant, Sofa, Television, Toilet}, are subset of the ones in PROCTHOR, we remove the parameters in the goal embedder corresponding to the rest of 10 objects presented in the pretraining stage.

## A.2 WHAT'S ENCODED IN THE CODEBOOK MODULE?

We expand our analysis of nearest neighbor retrieval to further evaluate the information captured within our codebook module. Utilizing a set of 10k frames from ProcTHOR, each accompanied by a corresponding goal specification that may or may not be visible in the frame, we input these frames and their associated goals into our pretrained EmbCLIP-Codebook model. This process generates hidden compact representations $h$, which are convex combinations of learnable codes $\{c_i\}_{i=1}^K$. To visually assess the information encoded in each individual learnable code $c_i$, we perform nearest neighbor retrieval on the dataset using each code $c_i$ as a query. Fig. 8 illustrates the 8 nearest neighbors for 9 different codes. The figure reveals that certain codes primarily contribute to goal visibility, determined by their proximity to the agent, while others capture various geometric and semantic details of scene layouts, such as walkable areas, corners, doors, etc.

Table 6: We fine-tune the CLIP ResNet visual encoder in an end-to-end manner for an additional 30M steps. The table shows the results for the frozen backbone and after end-to-end finetuning. EmbCLIP-Codebook consistently outperforms EmbCLIP both with frozen and fine-tuned visual encoder.

| Benchmark | Model | Object navigation | | |
|---|---|---|---|---|
| | | SR(%)↑ | EL↓ | SPL↑ |
| ProcTHOR-10k (validation) | EmbCLIP | 67.70 | 182.00 | 49.00 |
| | EmbCLIP-Codebook | 73.72 | 136.00 | 48.37 |
| | EmbCLIP + CLIP ResNet fine-tuning | 74.37 | 138.00 | 52.46 |
| | EmbCLIP-Codebook + CLIP ResNet fine-tuning | **79.38** | **120.00** | **52.67** |
| ARCHITECTHOR (0-shot) | EmbCLIP | 55.80 | 222.00 | 38.30 |
| | EmbCLIP-Codebook | 58.33 | 174.00 | 35.57 |
| | EmbCLIP + CLIP ResNet fine-tuning | 59.00 | 182.00 | **38.92** |
| | EmbCLIP-Codebook + CLIP ResNet fine-tuning | **62.58** | **168.00** | 37.10 |
| AI2-iTHOR (0-shot) | EmbCLIP | 70.00 | 121.00 | 57.10 |
| | EmbCLIP-Codebook | **78.40** | **86.00** | 54.39 |
| | EmbCLIP + CLIP ResNet fine-tuning | 72.53 | 88.00 | 59.28 |
| | EmbCLIP-Codebook + CLIP ResNet fine-tuning | 77.75 | 99.00 | **59.33** |

## A.3 COMPARISON WITH OTHER INFORMATION-BOTTLENECKED BASELINES

We ablate the choice of the bottleneck architecture and conduct a comparison between our codebook module and two alternative information-bottlenecked baselines. Specifically, we evaluate against **EmbCLIP-AE**, which employs an auto-encoder on the goal-conditioned embedding $E$. This autoencoder comprises a series of linear layers with ReLU activation functions, mapping the representation to $\mathcal{P} \in \mathcal{R}^{256}$ and $h \in \mathcal{R}^{10}$ before resizing back to $\hat{E} \in \mathcal{R}^{1574}$. Additionally, we introduce **EmbCLIP-SelfAttn** as another baseline, utilizing self-attention on the CLIP feature maps. To achieve this, we merge the compressed CLIP feature map $v$ with the goal embedding g, resulting in a $32 \times 7 \times 7$ goal-conditioned feature map. This map is then processed through the self-attention module, where 1x1 convolutions serve as the $k$, $q$, and $v$ functions. Due to limited computational resources and time constraints during the discussion period, the results are reported for the Object Navigation task at 210M training steps as shown in Table 5.

## A.4 WHAT ARE THE FAILURE CASES FOR EMBCLIP-CODEBOOK?

Fig. 9 illustrates examples of two modes of failure in our EmbCLIP-Codebook agent: perception and exploration. Although the codebook module enables the agent to effectively filter out distractions and concentrate on the target object, the agent's performance remains constrained by the perceptual capabilities of the pretrained visual encoder. The top row of Fig. 9 showcases examples of failures related to the perception. These instances predominantly occur when the goal object is either too small or challenging to identify (the baseball bat on the table in the top left example). In these scenarios, although the agent traverses past the object, it fails to accurately locate the target. The second row of the figure presents additional instances of failure, wherein the agent fails to explore specific areas of the environment where the target object is located.

## A.5 END-TO-END FINE-TUNING OF THE VISUAL ENCODER

In this section, we investigate the impact of end-to-end finetuning the entire policy model, including the CLIP ResNet visual encoder. Initially, we trained the models with the visual backbone frozen for 420 million steps, as depicted in Figure 2, with the results in Table 1. Subsequently, we fine-tuned the entire policy model, including the visual encoder, end-to-end for an additional 30 million steps. This process of fine-tuning the entire policy model incurs high computational costs. To facilitate this, we utilize multi-node training on 4 servers, each equipped with eight A-100-80GB GPUs. Moreover, we found that it is important to (1) employ a larger number of samplers (128) to match the substantial batch size (32,768) used in the CLIP's pretraining (Radford et al., 2021), and (2) use a small learning rate ($1e^{-6}$) for training stability. As shown in Table 6, while there is a substantial improvement from fine-tuning the entire policy model, including the visual encoder, compared to the frozen one, a noticeable gap still exists between fine-tuned EmbCLIP and fine-tuned EmbCLIP-Codebook. Remarkably, with further end-to-end finetuning, EmbCLIP-Codebook achieves significantly superior results compared to all other models.

Table 7: Adding a dropout on top the probability distribution over the latent codes acts as a codebook regularization and improves the performance. (Models are evaluated at 150M training steps.)

| Benchmark | Model | Object navigation | | |
|---|---|---|---|---|
| | | SR(%)↑ | EL↓ | SPL↑ |
| ProcTHOR-10k (validation) | EmbCLIP | 59.35 | **115.00** | 45.27 |
| | EmbCLIP-Codebook w/o Dropout | 62.52 | 126.00 | 45.46 |
| | EmbCLIP-Codebook | **71.39** | 147.00 | **47.46** |
| ARCHITECTHOR (0-shot) | EmbCLIP | 45.67 | **127.00** | 33.68 |
| | EmbCLIP-Codebook w/o Dropout | 48.08 | 164.00 | 33.61 |
| | EmbCLIP-Codebook | **57.17** | 192.00 | **35.65** |
| AI2-iTHOR (0-shot) | EmbCLIP | 58.93 | **67.00** | 51.55 |
| | EmbCLIP-Codebook w/o Dropout | 66.93 | 102.00 | 54.13 |
| | EmbCLIP-Codebook | **73.47** | 108.00 | **54.40** |

## A.6 CODEBOOK REGULARIZATION

Codebook collapse is a prevalent issue marked by the assignment of overconfident or high probabilities to a specific subset of latent codes in the codebook, causing the majority of codes to be underutilized. This restrains the models from fully leveraging the codebook's capacity. While it's important to note that our codebook module doesn't experience a complete collapse, incorporating regularization into the codebook enhances overall performance by promoting a more balanced utilization over all the latent codes. While various solutions have been suggested, we have found that applying a simple dropout on top of the codebook probabilities is the most effective and straightforward approach. In Fig. 10, we compare the average codebook probability distributions between two models—one trained with a $0.1$ dropout probability and the other without any dropout. The figure shows that this regularization leads to a more uniform average usage of the latent codes. Fig. 11 compares the training curves of the two models and Table 7 presents the corresponding evaluation results. Clearly, introducing such regularization to the codebook results in an improved performance. We additionally tried Linde-Buzo-Splitting algorithm (Linde et al., 1980) to balance the codebook useage. More specifically, during the training process, if there is a code vector that is underutilized, we perform a split on the most frequently used latent code, creating two equal embeddings and replacing the unused one. The corresponding training curve is depicted in Fig. 11, which shows a severe imbalance and eventual collapse during the training.

## A.7 COMPARISON OF AGENT'S BEHAVIOR

To enhance our understanding of the agent's navigation behavior, we delved deeper than the conventional embodied navigation metrics. We assessed the sequential actions and curvature values, offering empirical insights into the agent's behavior throughout the navigation episodes. Fig. 12 shows the action distribution of EmbCLIP-Codebook compared to EmbCLIP baseline in ARCHITECTHOR scenes. As shown in the plot, EmbCLIP-Codebook mostly favors *MoveAhead* actions while EmbCLIP baseline performs lots of redundant rotations. This observation is also supported by Fig. 13 which plots the frequencies of sequential *MoveAhead* actions taken by the agents. EmbCLIP-Codebook takes sequential *MoveAhead* actions much more frequently than the baseline resulting in smoother trajectories travelled by the agent. This provides further evidence for the lower average curvature for the EmbCLIP-Codebook trajectories as reported in Table 1. This is qualitatively visualized in the top-down maps shown in Fig. 14.

## A.8 COMPARISON TO OTHER BASELINES

Table 8 compares our method with 2 other baselines SGC (Singh et al., 2023) and NRC (Wallingford et al., 2023) on 3 Object Navigation benchmarks based on the numbers reported in the papers. Scene Graph Contrastive (SGC) improves the representation by building a scene graph from the gent's observations. Neural Radiance Field Codebooks (NRC) learns object-centric representations through novel view reconstruction and finetunes the representation for Object Navigation task. As shown in the table, we achieve better success rate and lower episode length compared to all methods across all 3 benchmarks.

Table 8: Comparisons against 2 other methods SGC (Singh et al., 2023) and NRC (Wallingford et al., 2023) on Object Navigation Benchmarks. Our method outperforms all the baselines on both Success Rate and Episode Length in all 3 benchmarks.

| Benchmark | Model | Object navigation | | |
|---|---|---|---|---|
| | | SR(%)↑ | EL↓ | SPL↑ |
| ARCHITECTHOR (0-shot) | SGC (Singh et al., 2023) | 53.80 | 204.00 | 34.80 |
| | EmbCLIP | 55.80 | 222.00 | **38.30** |
| | **EmbCLIP+Codebook** | **58.33** | **174.00** | 35.57 |
| RoboTHOR-Test (0-shot) | NRC (Wallingford et al., 2023) (finetuned) | 50.10 | - | 23.90 |
| | EmbCLIP (0-shot) | 51.32 | - | **24.24** |
| | **EmbCLIP+Codebook** (0-shot) | **55.00** | - | 23.65 |
| AI2-iTHOR (0-shot) | SGC (Singh et al., 2023) | 71.40 | 124.00 | **59.30** |
| | EmbCLIP | 70.00 | 121.00 | 57.10 |
| | **EmbCLIP+Codebook** | **78.40** | **86.00** | 54.39 |

Table 9: Ablations of Different Codebook Hyper-Parameters on 3 Object Navigation Benchmarks.

| Benchmark | Codebook Indexing | Codebook Size | Goal-Conditioned | *Object Navigation* | | |
|---|---|---|---|---|---|---|
| | | | | **SR(%)↑** | **EL↓** | **SPL↑** |
| ProcTHOR-10k (validation) | Softmax | 1024 × 10 | ✓ | 72.56 | 119 | 48.88 |
| | | 256 × 10 | ✓ | **73.4** | 124 | **49.84** |
| | | 256 × 10 | ✗ | 65.31 | 134 | 43.62 |
| | Top-1 gating | | | 38.06 | **110** | 29.39 |
| | Top-8 gating | 256 × 10 | ✓ | 68.22 | 160 | 45.86 |
| | Top-32 gating | | | 73.27 | 138 | 48.5 |
| | EmbCLIP Baseline | - | - | 59.81 | 176 | 43.91 |
| ARCHITECTHOR (0-shot) | Softmax | 1024 × 10 | ✓ | **60.00** | 160 | **37.33** |
| | | 256 × 10 | ✓ | 55.67 | 171 | 34.49 |
| | | 256 × 10 | ✗ | 48.42 | 181 | 31.02 |
| | Top-1 gating | | | 25.83 | **112** | 20.00 |
| | Top-8 gating | 256 × 10 | ✓ | 52.50 | 222 | 32.07 |
| | Top-32 gating | | | 59.33 | 188 | 36.14 |
| | EmbCLIP Baseline | - | - | 50.5 | 209 | 35.64 |
| AI2-iTHOR (0-shot) | Softmax | 1024 × 10 | ✓ | **76.4** | **65** | **57.4** |
| | | 256 × 10 | ✓ | 72.67 | 93 | 54.82 |
| | | 256 × 10 | ✗ | 66.93 | 107 | 48.53 |
| | Top-1 gating | | | 46.13 | 100 | 37.5 |
| | Top-8 gating | 256 × 10 | ✓ | 73.2 | 129 | 51.23 |
| | Top-32 gating | | | 72.93 | 101 | 50.57 |
| | EmbCLIP Baseline | - | - | 60.80 | 137 | 52.35 |

## A.9 CODEBOOK ABLATIONS

Table 9 compares different codebook hyperparameters including codebook size, scoring mechanism applied to the codebook entries and the choice of representation that undergoes compression by the codebook (goal-conditioned or not). All models are trained on PROCTHOR-10k houses for 300M training steps and evaluated zero-shot on ProcTHOR (validation), ArchitecTHOR and AI2-iTHOR benchmarks. We ablate 2 different codebook sizes specifically $256 \times 10$ and $1024 \times 10$. To underscore the significance of goal-conditioning, we employ the codebook to compress representations both before and after they are fused with the goal ($v$ or $E$ embeddings). As shown in the table, our codebook bottleneck is only effective when the visual representation is conditioned on the goal across all the benchmarks. Moreover, to further intensify the bottlenecking mechanism, we selectively retain only the top-$N$ scores, disregarding the rest, before computing the convex combination of the codebook entries. This is inspired by the gating mechanism in Mixture of Experts (Riquelme et al., 2021; Shazeer et al., 2017) and results in only the top $N$ most important codes to contribute to the bottlnecked representation at any given time. As shown in the table, selecting the top 32 codes for the convex combination can sometimes work as well as using all 256 latent codes. Using a bigger codebook size (1024 instead of 256) improves the zero-shot generalization in some of the benchmarks.

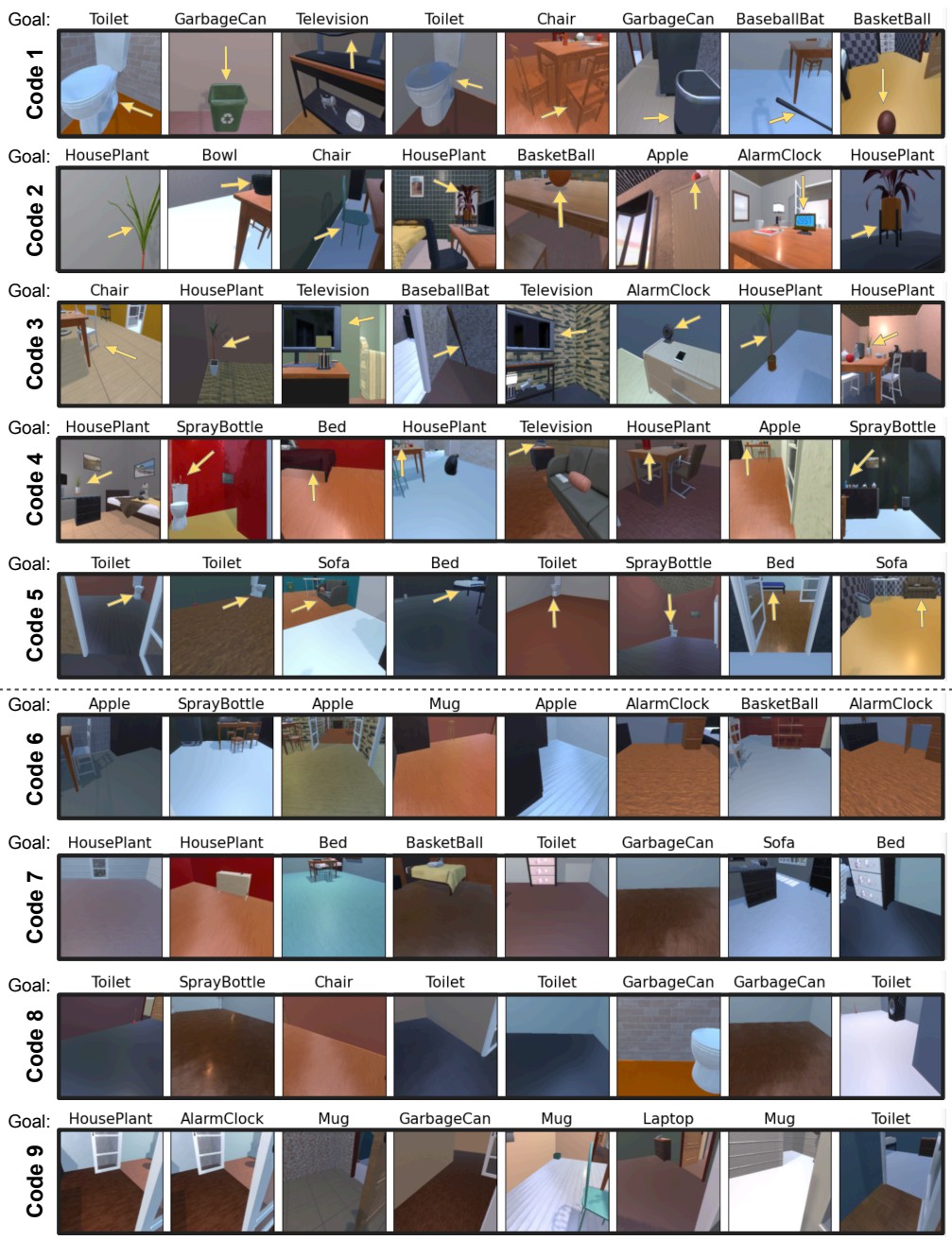

Figure 8: **Nearest-neighbor retrieval in the hidden compact Representation Space** $h \in \mathcal{R}^{10}$ **using $\{c_i\}_{i=1}^{K}$ as queries**. We show the 8 nearest neighbors of 9 learnable codes in the codebook module. While some codes seem to be responsible for encoding the object goal visibility depending on different proximities to the agent (codes 1 to 5), some others encode other semantic and geometric information about the scene layout such as walkable surfaces (code 6,7), corners (code 8), and doors (code 9) etc.

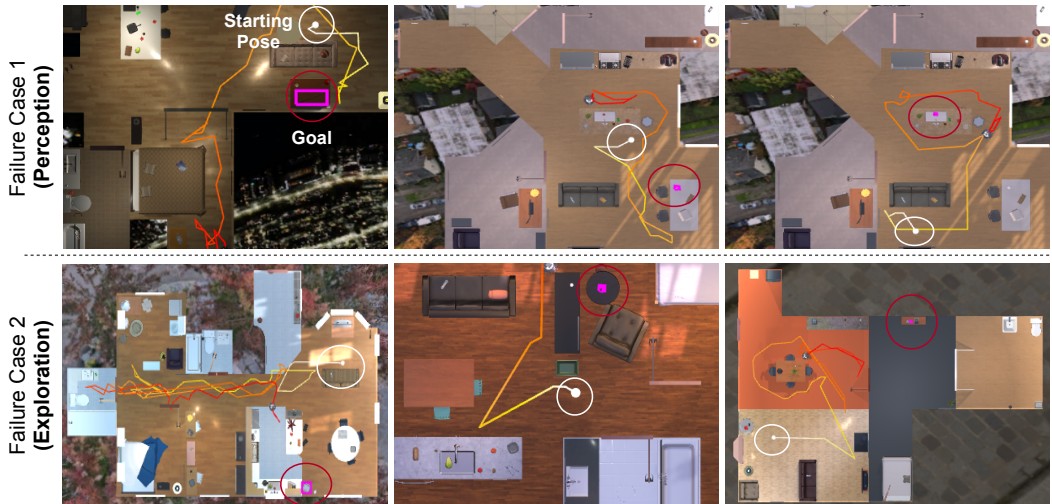

Figure 9: **Failure cases of EmbCLIP-Codebook in Object Navigation**. Top row shows the failure examples related to perception, where the agent traverses past the goal object but fails to identify it. The bottom row shows another failure mode where the agent fails to explore specific areas of the environment where the target object is located. (Agent's starting pose is shown in white circle and the target object is shown in a red circle)

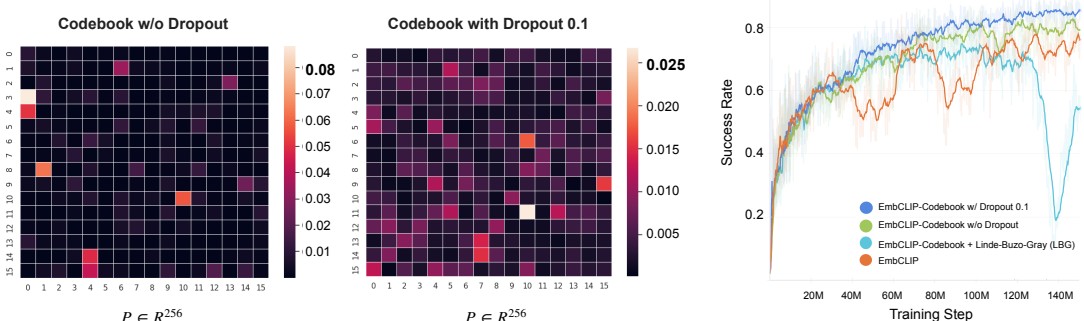

Figure 10: Average probability distribution $\mathcal{P} \in \mathcal{R}^{256}$ over the latent codes comparing the codebook trained with 0.1 dropout probability and without any dropout. Using dropout clearly results in a more uniform utilization of all the latent codes in the codebook module. ($\mathcal{P}$ is reshaped to $16 \times 16$ for visualization.)

Figure 11: Success Rate training curve comparing EmbCLIP (baseline), EmbCLIP-Codebook with and without dropout regularization, and EmbCLIP-Codebook trained with Linde-Buzo-Gray splitting algorithm.

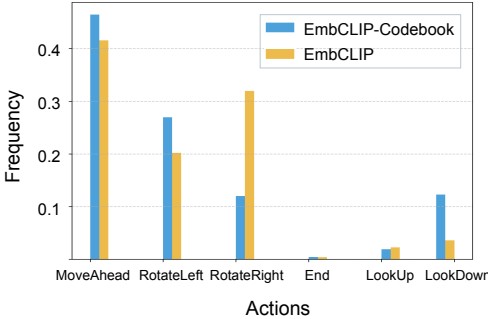

Figure 12: Action frequencies in ArchitecTHOR validation trajectories.

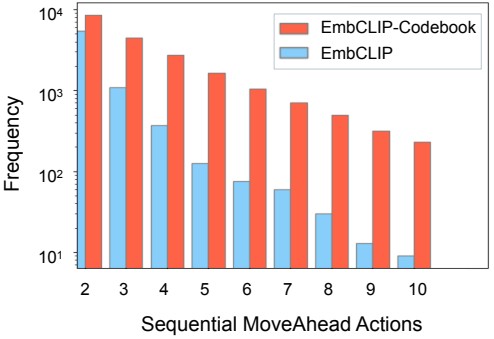

Figure 13: Frequencies of sequential *MoveAhead* actions in ArchitecTHOR validation trajectories.

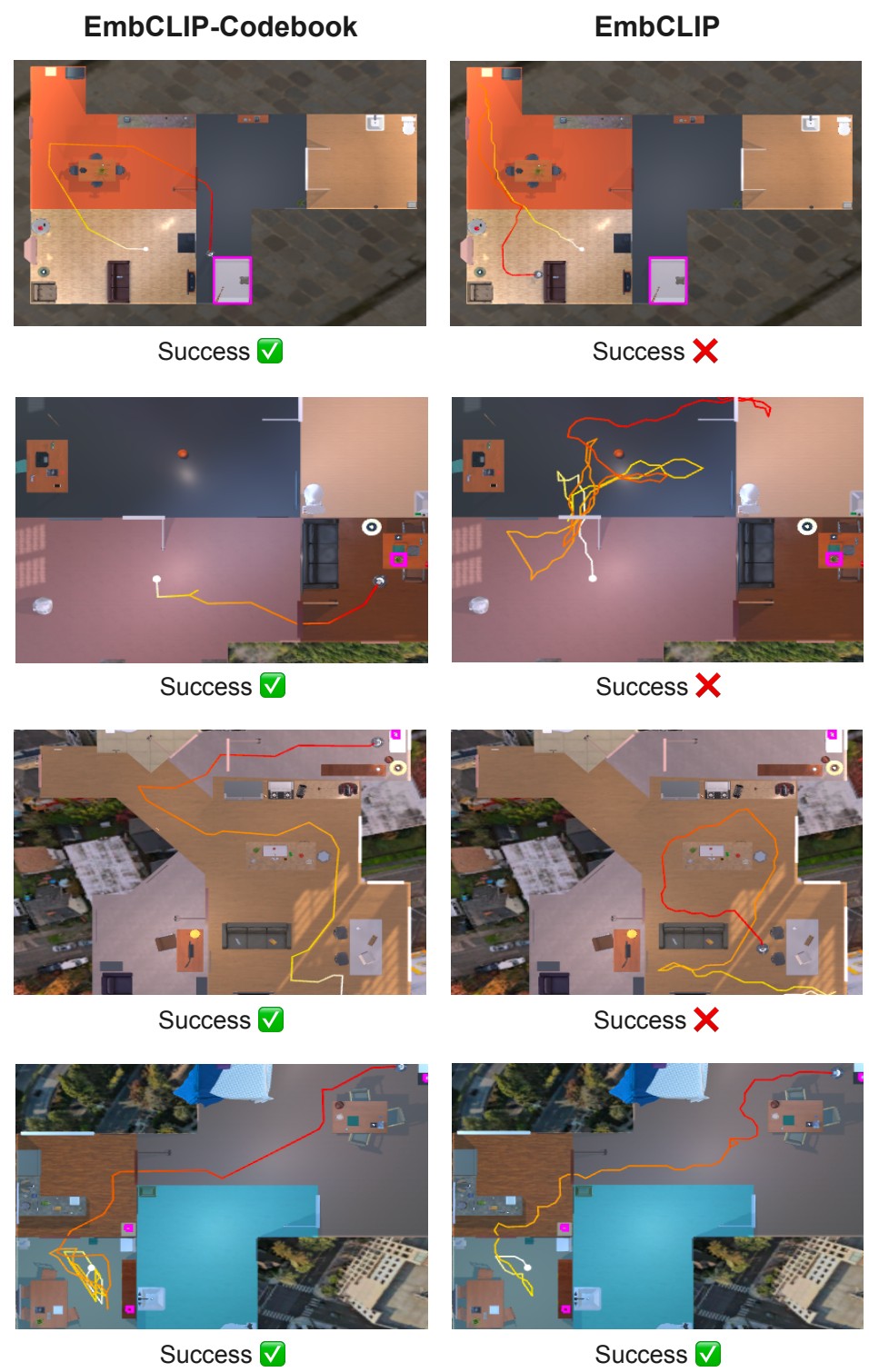

Figure 14: Top-down Maps of Agents' Trajectories. Our agent travels in smoother paths and explores the environment more effectively.

