# OpenReview forum: "Selective Visual Representations Improve Convergence and Generalization for Embodied AI"
_ICLR.cc/2024/Conference — ICLR 2024 spotlight_

### Official Review · Reviewer_NmEd · 2023-11-01

**Soundness:** 3 good
**Presentation:** 3 good
**Contribution:** 3 good
**Rating:** 6
**Confidence:** 4

**Summary:**

This work starts from a very attractive motivation: redundant information will blind people to making correct actions. The authors utilize a simple yet effective trick, i.e., adding a parameter-efficient codebook, and expecting it to filter out the unnecessary information in the embeddings. The method significantly boosts the performance of the baseline EmbCLIP on several benchmarks.

**Strengths:**

I am quite appealed by the introduction, which I believe illustrates a very interesting motivation and will inspire the community. The method is also very simple and effective. Simply incorporating a codebook module is able to improve the baseline EmbCLIP by a large margin on several benchmarks. Overall, I believe this paper makes a good contribution, especially in terms of motivations and methods.

**Weaknesses:**

One of the weaknesses is also related to its strengths, i.e., motivation. I was quite attracted by the example of Figure 1, in which the authors describe the situation that the keys would only lie on a flattened surface, not a corner or somewhere else. I do like the motivation and expect the authors to incorporate such a human prior into the method yet I did not find it in the paper. I would suggest the authors consider incorporating a text prompt on the CLIP text model such as "the chair is usually on the floor" etc. Not necessarily for all the benchmarks but I think it would be interesting to see whether several cases can be improved.

The second weakness is about codebook collapse, this is a good motivation and interesting problem that is also related to interpreting what codebook really learns. I would expect whether the current method exists codebook collapse without dropout during training. This means the authors may need to conduct ablation studies and provide illustrations about collapse or not.

**Questions:**

Should adding a skip connection between \hat{E} and E help improve the current version, i.e., \hat{E} = theta(E)+Code Module(E)? I think bottleneck representations such as ResNet etc will usually do so.

For the other questions please see weakness.

---

> ### Author Response · Authors · 2023-11-22
>
> We thank you for your valuable feedback. In the following, we use W for Weaknesses and Q for Questions.
>
>
> **W1) Incorporating text prompts as a human prior into the models**
>
> That’s a very interesting suggestion! It can definitely be applied as a future work of our approach. In our current proposed method, we anticipate the automatic extraction of such common-sense knowledge and human priors from the observations through the bottleneck. Integrating such priors in the form of language could indeed be a compelling direction for future work.
>
>
> **W2) Codebook collapse ablations**
>
> Thanks for the insightful comment. We included an ablation study in section **A.7**. Codebook collapse is a prevalent issue defined as the assignment of high probabilities to a specific subset of latent codes in the codebook, causing the majority of codes to be underutilized. While it’s important to note that our codebook module doesn’t experience a complete collapse, incorporating regularization into the codebook enhances overall performance by promoting a more balanced utilization of all the latent codes. While various solutions have been suggested, we have determined that applying a simple dropout on top of the codebook probabilities is the most effective and straightforward approach. In **Fig. 10**, we compare the average codebook probability distributions between two models—one trained with a 0.1 dropout probability and the other without any dropout. The figure shows that this regularization leads to a more uniform average usage of the latent codes. **Table 8** presents the corresponding evaluation results. Clearly, introducing such regularization to the codebook module results in an improved performance. We additionally tried the **Linde-Buzo-Gray[1]** splitting algorithm as an alternative approach to balance the codebook usage. More specifically, during the training process, if there is a code vector that is underutilized, we perform a split on the most frequently used latent code, creating two equal embeddings and replacing the unused one. The corresponding training curve is depicted in **Fig. 11**, which shows a severe imbalance and eventual collapse during the training.
>
> [1] Yoseph Linde, Andres Buzo, and Robert Gray. An algorithm for vector quantizer design. In IEEE Transactions on communications, 1980.
>
>
> **Q1) What’s the effect of adding a skip connection?**
>
> We observed that incorporating a skip connection into our architecture leads to poorer performance, likely due to overfitting. There is a possibility that our existing hyperparameters may need adjustment to better suit the architecture with the skip connection, but despite our efforts, we were unable to identify the optimal hyperparameters, given the time constraints of the discussion period. We believe this requires further investigation.  Nonetheless, based on our current findings, it appears that employing a full bottleneck yields better results than introducing a skip connection. Table below shows the results for 300 million steps of training.
>
> | **Benchmark**             | **Model**                          | **SR(%)↑** | **EL↓** | **SPL↑** |
> | ------------------------- | ---------------------------------- | ---------- | ------- | -------- |
> | **ProcTHOR-10k (test)**   | EmbCLIP-Codebook                   | 73.40      | 124.00  | 49.84    |
> |                           | EmbCLIP-Codebook + Skip Connection | 44.58      | 170.00  | 26.11    |
> | **ArchitecTHOR (0-shot)** | EmbCLIP-Codebook                   | 55.67      | 171.00  | 34.49    |
> |                           | EmbCLIP-Codebook + Skip Connection | 36.80      | 169.00  | 21.20    |
> | **AI2-iTHOR (0-shot)**    | EmbCLIP-Codebook                   | 72.67      | 93.00   | 54.82    |
> |                           | EmbCLIP-Codebook + Skip Connection | 49.60      | 119.00   | 33.75    |

---

> > ### Comment · Reviewer_NmEd · 2023-11-23
> > **Response**
> >
> > Thanks for the reply. The authors address my main concerns about collapse and skip connection. It is a good paper but I will keep the score because I wish the authors could address my largest concern about integrating human priors, which reflects the motivation of the paper.

---

### Official Review · Reviewer_vxKH · 2023-11-01

**Soundness:** 3 good
**Presentation:** 4 excellent
**Contribution:** 3 good
**Rating:** 8
**Confidence:** 3

**Summary:**

Visual encoders like CLIP capture general purpose scene information which includes details not relevant to the task. The paper proposes to learn a codebook module that selectively filters information from visual representations specific for a task. They demonstrate the approach on a wide range of Embodied AI tasks across several large-scale benchmarks. Additionally, they present qualitative analysis showing that the codebook module better encodes task information which lends to more robust navigational policies.

**Strengths:**

Approach is simple, easy to follow, and well motivated.

Codebook representations and policy are learned end-to-end.

Results are shown across a diverse set of tasks and benchmarks. Improvement over EmbCLIP seems significant. Also report a comprehensive set of metrics including a new metric Success Weighted by Episode Length which accounts for actions like changing viewing angles that also requires time and effort.

Interesting experiments showing that with minimal finetuning of the Adaptation Module with a frozen codebook can help transfer to new visual domains.

Saliency visualizations are informative of where the model is paying attention to and highlights relevant tasks objects.

Nice set of ablation studies on codebook latent size and linear probes.

**Weaknesses:**

Does the codebook work for other types of general purpose visual encoders? This work seems to only show the codebook module applied to EmbCLIP.

**Questions:**

Could the codebook be learning to localize the goal information in the scene? The input to the codebook contains both the visual observation and a language description of the goal. It could be possible that the codebook is simply acting as a bridge between the two input modalities.

What are some of the failure cases of the EmbCLIP + codebook method? Are there instances where codebook fails to identify task objects?

Are there cases where two objects of the same class are present in the scene, but the task requires the model to pick one of the objects.

---

> ### Author Response · Authors · 2023-11-22
>
> We thank you for your valuable feedback. In the following, we use W for Weaknesses and Q for Questions.
>
>
> **W1) Applying the codebook to other pretrained visual encoders**
>
> Thanks a lot for your great comment! We included section **A.2** in the appendix which shows the codebook module is representation-agnostic and can be applicable to other pretrained visual encoders as well. We used pretrained **DINOv2**[1] visual features and used the codebook to bottleneck the new goal-conditioned representations. We use the frozen **DINOv2 ViT-S/14** model to encode RGB images into a 384x7x7 tensor. We fuse this tensor with a 32-dimensional goal embedding and the previous action embedding and flatten the result to obtain a 1574-dimensional goal-conditioned observation embedding. We employed a codebook with similar dimensions, K = 256 and Dc = 10, to bottleneck the goal-conditioned representations. As shown in **Table 5** (shown in below as well), our approach outperforms the DINOv2 baseline models across a variety of Object Navigation metrics in various benchmarks. This experiment underscores the capability of our codebook module in effectively bottlenecking other visual features for embodied-AI tasks.
>
> | Benchmark                   | Model              | SR(%)↑ | EL↓   | Curvature↓ | SPL↑  | SEL↑  |
> |-----------------------------|--------------------|-------:|------:|-----------:|------:|------:|
> | **ProcTHOR-10k (test)**     | DINOv2             | 74.25  | 151.00| 0.24       | 49.53 | 43.20 |
> |                             | +Codebook (Ours)   | **76.31** | **129.00**| **0.12**   | **50.26** | **44.70** |
> | **ArchitecTHOR (0-shot)**   | DINOv2             | 57.25  | 218.00| 0.25       | 36.83 | 29.00 |
> |                             | +Codebook (Ours)   | **59.75** | **194.00**| **0.11**   | **36.00** | **31.70** |
> | **AI2-iTHOR (0-shot)**      | DINOv2             | 74.67  | 97.00 | 0.19       | 59.45 | 26.50 |
> |                             | +Codebook (Ours)   | **76.93** | **68.00** | **0.07**   | **60.14** | **28.30** |
> | **RoboTHOR (0-shot)**       | DINOv2             | 60.54  | -     | -          | **29.36** | -     |
> |                             | +Codebook (Ours)   | **61.03** | -     | -          | 28.01 | -     |
>
> [1] Maxime Oquab, Timothée Darcet, Théo Moutakanni, Huy Vo, Marc Szafraniec, Vasil Khalidov, Pierre Fernandez, Daniel Haziza, Francisco Massa, Alaaeldin El-Nouby, et al. Dinov2: Learning robust visual features without supervision. arXiv preprint arXiv:2304.07193, 2023
>
> **Q1) Codebook as a bridge between modalities?**
>
>
> Yes, our model encodes goal related information beyond just the appearance of the goal (location, context, places, etc). Our model bridges between task-related information and visual information. Our tasks are represented as an embedding and our codebook connects the two. Figure 8 shows the encoded information in different latent codes. While some codes are directly responsible for goal-related information such as goal visibility and proximity to the agent, others encode other information required for the object navigation task such as walkable areas, corners, etc.
>
>
> **Q2) What are the failure cases of the EmbCLIP-Codebook?**
>
>
> We included section **A.5** which analyzes the failure cases for EmbCLIP-Codebook. **Fig. 9** illustrates examples of two modes of failure in our EmbCLIP-Codebook agent: perception and exploration. Although the codebook module enables the agent to effectively filter out distractions and concentrate on the target object, the agent’s performance remains constrained by the perceptual capabilities of the pretrained visual encoder. So there exists instances where the codebook fails to identify target objects.
> The top row of **Fig. 9** showcases examples of failures related to perception. These instances predominantly occur when the goal object is either too small or challenging to identify (the baseball bat on the table in the top left example). In these scenarios, although the agent traverses past the object, it fails to accurately locate the target. The second row of the figure presents additional instances of failure, wherein the agent fails to explore specific areas of the environment where the target object is located.
>
>
>
> **Q3) Are there cases where two objects of the same class are present in the scene?**
>
>
> Yes, that is indeed possible. Take, for example, a scenario where the target object is a 'chair.' In a typical scene, multiple chairs are often found around dining tables. The objective of Object Goal Navigation is to find any one instance of the target object type – in this case, any chair in the scene. Therefore, the task requires the agent to find any single instance of the specified object type.

---

### Official Review · Reviewer_YjyH · 2023-11-01

**Soundness:** 3 good
**Presentation:** 3 good
**Contribution:** 2 fair
**Rating:** 8
**Confidence:** 3

**Summary:**

The paper presents a parameter efficient approach to filter out task-irrelevant information encoded by visual encoders, such as CLIP, in embodied AI tasks. This approach leverages a compact, learnable codebook module to establish a task-specific filter for visual observations. The codebook is trained to optimize task performance, serving as a filter that directs the agent's attention toward task-relevant visual cues. The experimental results demonstrate performance improvement in object goal navigation and object displacement tasks across various benchmarks. Qualitative analysis illustrates that agents become more proficient in exploring their surroundings, retaining task-relevant information, and disregarding irrelevant visual details.

**Strengths:**

1) The paper proposes a simple parameter efficient approach for adapting representations which leads to consistent performance improvement on multiple benchmarks.
2) Because the model learns to ignore the irrelevant parts of the visual inputs, it is able to do more efficient exploration and navigation..
3) The proposed finetuning approach of the codebook module is clever.

**Weaknesses:**

1) Given that the proposed approach is about the codebook module, it would have been good to see if this approach could be applied to other pretrained visual encoders, for eg ViT based models.
2) It's not clear how their approach compares against a full-scale visual encoder fine tuning baseline? Finetuning of the visual encoder should also allow it to forget information that is not relevant to the task. While finetuning is computationally expensive, it will still be interesting to see how EmbCLIP-codebook performs with respect to it.
3) It is also not clear how important is the use of the codebook vectors compared to just the introduction of a bottleneck in the architecture? To test this, I recommend the authors try using their proposed bottleneck architecture without the codebook vectors.

**Questions:**

I have listed my main concerns in the weaknesses section. I will be happy to increase the score if the authors can answer those with relevant experiments. Other than that, I have some other questions and suggestions that I list below:
1) What is meant by “Samplers”, which is mentioned in the experimental details?
2) Why is RobotTHOR missing some metrics in Table 1?
Typos:
Page 6: HMD semantics -> HM3D Semantics. Also missing citation for HM3D Semantics
Page 5: temperture -> temperature
Page 2: codebook better encodes -> codebook encodes better
Page 17: gent’s -> agent’s

---

> ### Author Response · Authors · 2023-11-22
>
> We thank you for your valuable feedback. In the following, we use W for Weaknesses and Q for Questions.
>
> **W1) Applying the codebook to other pretrained visual encoders**
>
>
> Thanks for the great feedback! We added section **A.2** in the appendix. In order to show the applicability of the codebook to other visual encoders, we used pretrained **DINOv2**[1] visual features and applied the codebook to bottleneck the new goal-conditioned representations. We use the frozen **DINOv2 ViT-S/14** model to encode RGB images into a 384x7x7 tensor. We fuse this tensor with a 32-dimensional goal embedding and the previous action embedding and flatten the result to obtain a 1574-dimensional goal-conditioned observation embedding. We employed a codebook with similar dimensions, K = 256 and Dc = 10, to bottleneck the goal-conditioned representations. As shown in **Table 5** (shown in below as well), our approach outperforms the DINOv2 baseline models across a variety of Object Navigation metrics in various benchmarks. This experiment underscores the capability of our codebook module in effectively bottlenecking other visual features for embodied-AI tasks.
>
> | Benchmark                   | Model              | SR(%)↑ | EL↓   | Curvature↓ | SPL↑  | SEL↑  |
> |-----------------------------|--------------------|-------:|------:|-----------:|------:|------:|
> | **ProcTHOR-10k (test)**     | DINOv2             | 74.25  | 151.00| 0.24       | 49.53 | 43.20 |
> |                             | +Codebook (Ours)   | **76.31** | **129.00**| **0.12**   | **50.26** | **44.70** |
> | **ArchitecTHOR (0-shot)**   | DINOv2             | 57.25  | 218.00| 0.25       | 36.83 | 29.00 |
> |                             | +Codebook (Ours)   | **59.75** | **194.00**| **0.11**   | **36.00** | **31.70** |
> | **AI2-iTHOR (0-shot)**      | DINOv2             | 74.67  | 97.00 | 0.19       | 59.45 | 26.50 |
> |                             | +Codebook (Ours)   | **76.93** | **68.00** | **0.07**   | **60.14** | **28.30** |
> | **RoboTHOR (0-shot)**       | DINOv2             | 60.54  | -     | -          | **29.36** | -     |
> |                             | +Codebook (Ours)   | **61.03** | -     | -          | 28.01 | -     |
>
>
> [1] Maxime Oquab, Timothée Darcet, Théo Moutakanni, Huy Vo, Marc Szafraniec, Vasil Khalidov, Pierre Fernandez, Daniel Haziza, Francisco Massa, Alaaeldin El-Nouby, et al. Dinov2: Learning robust visual features without supervision. arXiv preprint arXiv:2304.07193, 2023

---

> ### Author Response · Authors · 2023-11-22
>
> **W2) Full-scale visual encoder fine-tuning as a baseline**
>
>
> Thank you for the suggestion. To explore the impact of full-scale fine-tuning, we fine-tuned the entire policy, including the visual encoder, for both EmbCLIP and EmbCLIP-Codebook models. We used the best checkpoints reported in **Table 1** and fine-tuned the entire policy, including the ResNet-50 visual encoder, for an additional 30 million steps. Details of the implementation, including multi-node training, large batch size, and small learning rate, can be found in **Appendix 6**. The results are presented in the following table (also included in the updated paper as **Table 7**).
>
> | Benchmark                   | Model                                    | SR(%)↑ | EL↓ | SPL↑ |
> |-----------------------------|------------------------------------------|-------:|----:|-----:|
> | **ProcTHOR-10k (test)**     | EmbCLIP                                  | 67.70  | 182.00 | 49.00 |
> |                             | EmbCLIP-Codebook                         | 73.72  | 136.00 | 48.37 |
> |                             | EmbCLIP + CLIP ResNet fine-tuning        | 74.37  | 138.00 | 52.46 |
> |                             | EmbCLIP-Codebook + CLIP ResNet fine-tuning | **79.38** | **120.00** | **52.67** |
> | **ArchitecTHOR (0-shot)**   | EmbCLIP                                  | 55.80  | 222.00 | 38.30 |
> |                             | EmbCLIP-Codebook                         | 58.33  | 174.00 | 35.57 |
> |                             | EmbCLIP + CLIP ResNet fine-tuning        | 59.00  | 182.00 | **38.92** |
> |                             | EmbCLIP-Codebook + CLIP ResNet fine-tuning | **62.58** | **168.00** | 37.10 |
> | **AI2-iTHOR (0-shot)**      | EmbCLIP                                  | 70.00  | 121.00 | 57.10 |
> |                             | EmbCLIP-Codebook                         | **78.40** | **86.00** | 54.39 |
> |                             | EmbCLIP + CLIP ResNet fine-tuning        | 72.53  | 88.00 | 59.28 |
> |                             | EmbCLIP-Codebook + CLIP ResNet fine-tuning | 77.75 | 99.00 | **59.33** |
>
> As the table shows, while there is a substantial improvement from fine-tuning the entire policy model, including the visual encoder, compared to using the frozen one, a noticeable gap remains between the fine-tuned EmbCLIP and fine-tuned EmbCLIP-Codebook. Notably, with further end-to-end fine-tuning, EmbCLIP-Codebook achieves significantly superior results compared to all other models.
>
>
> **W3) Bottlenecked architecture without the codebook module as a baseline**
>
>
> Thanks for the insightful comment. We included a new section, **A.4**, in the appendix. **Table 6** compares our proposed bottleneck architecture (EmbCLIP-Codebook) with 2 other information bottleneck baselines on 3 Object Navigation benchmarks. **EmbCLIP-AE** utilizes an auto-encoder on the goal-conditioned embedding E. This autoencoder comprises a series of linear layers, mapping the representation to $\mathcal{P} \in \mathcal{R}^{256}$ and $h \in \mathcal{R}^{10}$ before resizing back to $\hat{E} \in \mathcal{R}^{1574}$. The results in Table 6 demonstrate the superior performance of our method compared to EmbCLIP-AE across all 3 benchmarks which underscores the importance of the codebook module in the proposed architecture.
>
> **Q1) What is meant by “Samplers” mentioned in experiment details?**
>
> The concept of a Sampler is defined in AllenAct [1]. The primary function of a Sampler is to sample an episode (or a Task) for the agent. Each episode specifies the action space, success criteria, and the rewards that the environment returns to the agent. After the agent completes an episode, the Sampler selects a new episode for the agent. Consequently, a greater number of Samplers indicates that a higher number of episodes are being processed by the agent in parallel. In other words, more Samples imply a larger batch size for training a policy model.
>
>
> [1] Luca Weihs and Jordi Salvador and Klemen Kotar and Unnat Jain and Kuo-Hao Zeng and Roozbeh Mottaghi and Aniruddha Kembhavi, “AllenAct: A Framework for Embodied AI Research”, Arxiv 2020, url: https://allenact.org/.
>
> **Q2) Why is RoboTHOR missing some metrics?**
>
>
> RoboTHOR test results are only available through submissions to the **leaderboard** ([link](https://leaderboard.allenai.org/robothor_objectnav/submission/ckaa7vjb33j953fg5h0g)). We also included the link to the submission in **a footnote on page 6**. The leaderboard metrics only include **Success Rate** and **SPL**.

---

### Official Review · Reviewer_X3Z2 · 2023-11-01

**Soundness:** 3 good
**Presentation:** 3 good
**Contribution:** 2 fair
**Rating:** 8
**Confidence:** 4

**Summary:**

This paper proposes an adaptive feature modulation module for embodied AI.  The core idea is to create a task-conditioned representation bottleneck, based on a learnable codebook, to select useful features for each robotic task.  The codebook is jointly trained to optimize the task reward.  This paper demonstrates superior efficacy of the proposed method, and conduct thorough ablation study / visualization of the proposed method.

**Strengths:**

1. Good ablation study.  Table 3 clearly shows the selected information of the codebook. Grad-CAM visualization also makes sense.

2. The paper is well written.  It's easy to follow especially that readers are primed with some high-level ideas from human vision.

3. The performance gain is significant compared to the baseline.

**Weaknesses:**

1. Learning codebook as information bottleneck helps discriminative mapping of observations to task outputs.  Meanwhile, throwing away information means that there's little knowledge shared among different task.  It'd be interesting to see if the performance comparison on zero-shot/few-shot learning of novel tasks.  My guess is that EmbCLIP would adapt toe the novel task faster and better than the proposed codebook method.


2. The paper does not provide comparison to other bottleneck-based baselines.  For example, one can learn a self-attention modules atop CLIP feature maps.  Also, learning an auto-encoder, where the bottleneck is low-dimensional latent feature, seems to be another reasonable baseline.  Both self-attention modules and the auto-encoder will condition on the goal task description and previous action.

**Questions:**

1. What's the performance of fine-tuning entire model on Habitat and adaptation module on HMD semantics (table 2)?  Also, what's the before-finetuning results of EmbCLIP+codebook on both benchmark?  Right now the comparison is conducted over different dataset, not apple-to-apple.

2. The nearest neighbor probably should be done by treating learned codes as query.  In stead of using the pooled/weighted sum features, the authors can use a one-hot vector to select each code and upsample it to 1568 dimension.  It'd be more interesting to see what are learned in the codebook and what do the retrievals look like for each task.

3. Perhaps the authors should compare with other information-bottleneck baselines, e.g. self-attention / autoencoder.  Such results could clarify if codebook / information bottleneck is the key.

---

> ### Author Response · Authors · 2023-11-22
>
> We thank you for the valuable feedback. In the following, we use W for Weaknesses and Q for Questions.
>
>
> **W1) Throwing away information using the codebook might result in worse performance in zero-shot/few-shot learning of novel tasks**
>
> Bottlenecking and discarding information does not necessarily lead to poorer generalization in novel scenarios. An illustrative example of this is found in regularization techniques such as dropout, wherein they effectively prevent overfitting, resulting in simpler and more generalizable models. Furthermore, our codebook module is specifically designed to be **task-specific**. Our main goal is to bottleneck the information to best suit the task in hand. Therefore, performing a new task requires training another task-specific codebook module. It is however expected of our proposed approach to be more generalizable when applied to the **same task in new domains**.
> As demonstrated in **Section 4.2** of the paper, our method significantly outperforms non-bottlenecked architectures in adapting to new visual domains through the lightweight fine-tuning of a small adaptation module.
>
>
> **W2, Q3) Comparison with other information bottleneck baselines**
>
> Thanks a lot for this insightful feedback. We included a new section, **A.4**, in the appendix. **Table 6** compares our proposed bottleneck architecture (EmbCLIP-Codebook) with 2 other information bottleneck baselines on 3 Object Navigation benchmarks. **EmbCLIP-AE** utilizes an auto-encoder on the goal-conditioned embedding E, while **EmbCLIP-SelfAttn** applies self-attention to the goal-conditioned CLIP feature maps. The results in Table 6 demonstrate the superior performance of our method compared to the other information bottleneck baselines across all 3 benchmarks.
>
>
>
> **Q1) Habitat zero-shot results and fine-tuning results on the same benchmark**
>
> Sorry for the confusion. All results are originally evaluated on the same benchmark (HM3D Semantics dataset) used in Habitat 2022 ObjectNav Challenge. We have updated **Table 2** to clarify our fine-tuning results.
> We’re also reporting the zero-shot results for both models here. The models are evaluated at 220M and 420M training steps. We’re not including the results in the paper as the 0-shot results are very low compared to the fine-tuned versions.
>
>
>
>
> | Benchmark              | Model            | Training Step | SR(%)↑ | EL↓ | SPL↑      |
> | ---------------------- | ---------------- | ------------- | ------------ | -------------- | -------- |
> |**Habitat Challenge 2022** (zero-shot) | EmbCLIP          | 220M          | **11.20**    | 313            | **6.50** |
> | **Habitat Challenge 2022** (zero-shot) | EmbCLIP-Codebook | 220M          | 8.15         | **253**        | 4.71     |
> |**Habitat Challenge 2022** (zero-shot) | EmbCLIP          | 420M          | 8.50         | 362            | 4.85     |
> |**Habitat Challenge 2022** (zero-shot) | EmbCLIP-Codebook | 420M          | **9.85**     | **345**        | **5.23** |
>
>
>
> **Q2) Nearest-neighbor analysis using learned codes as queries**
>
> Thanks for the interesting suggestion! We added this analysis in section **A.3** in the appendix. Using a set of 10k ProcTHOR frames each accompanied by a corresponding goal specification that may or may not be visible in the frame, we input these frames and their associated goals into our pretrained EmbCLIP-Codebook model. We save their corresponding 10-dim hidden compact representations h which are convex combinations of the latent codes. To visually assess the information encoded in each individual learnable code, we perform nearest neighbor retrieval on the dataset using each 10-dim codebook entries as queries. **Fig. 8** illustrates the 8 nearest neighbors for 9 different codes. The figure reveals that certain codes primarily contribute to goal visibility, determined by their proximity to the agent, while others capture various geometric and semantic details of scene layouts, such as walkable areas, corners, doors, etc.

---

> > ### Comment · Reviewer_X3Z2 · 2023-11-22
> >
> > Thanks for the response.  My questions are mostly answered and I'm happy to increase my ratings.

---

### Author Response · Authors · 2023-11-22
**To All Reviewers**

We thank the reviewers for their valuable feedback and are pleased that all reviewers recognize the effectiveness of our proposed method, noting its state-of-the-art results and great generalization ability across different benchmarks. We also thank the comments from reviewers _YjyH_, _NmEd_ and _vxKH_ regarding the simplicity, ease of understanding, and strong motivation behind our method. Reviewers _X3Z2_ and _vxKH_ have highlighted that our detailed ablation studies are informative and clearly illustrate the encoded visual cues in the proposed codebook. Additionally, reviewers _X3Z2_ and _NmEd_ have noted that our paper is well-written and is likely to inspire the community. Finally, reviewers _X3Z2_ and  _vxKH_ mention that saliency visualizations by GradCam are informative.

After carefully reading all the reviews, we have added the following updates:

- In response to suggestions from reviewers _YjyH_ and _vxKH_, we have replaced the ResNet-50, pretrained by CLIP, with DINOv2, and demonstrate that the model with our proposed codebook module continues to outperform the model without it. Details can be found in Appendix 2.
- In Appendix 3, we extended our nearest neighbor retrieval study using individual latent codes as queries in order to analyze the information captured in single codebook rows.
- Following recommendations from reviewers _X3Z2_ and _YjyH_, we compare our method against other information bottleneck baselines: (1) EmbCLIP-AE: replacing the codebook module with a simple autoencoder and (2) EmbCLIP-SelfAttn: using a self-attention layer to bottleneck the visual representation. Results and implementation details are included in Appendix 4. The results demonstrate the superior performance of our method compared to the other information bottleneck baselines across all 3 benchmarks.
- Appendix 5 presents qualitative examples of failure cases, along with a preliminary study of these cases.
- Appendix 6 explores the impact of end-to-end fine-tuning of the entire policy model, including the ResNet-50 visual encoder, pretrained by CLIP. With further end-to-end fine-tuning, EmbCLIP-Codebook achieves significantly superior results compared to all other models, including the end-to-end fine-tuned EmbCLIP baseline.
- In Appendix 7, we perform an ablation study on different techniques for codebook regularization.

All changes are available in the revised PDF and are highlighted in red for easy identification.

Please refer to our responses below for specific questions.

---

### Meta-Review · Area_Chair_bn1Y · 2023-12-09

**Metareview:**

**Summary:**
The paper introduces an approach in embodied AI for enhancing task-specific visual representations. The method uses a learnable codebook module as a task-conditioned bottleneck, filtering out irrelevant visual stimuli and optimizing task performance. Demonstrating significant improvements in object navigation and displacement tasks across multiple benchmarks, the approach also shows better adaptability to new environments and efficient exploration, focusing on task-relevant cues while discarding non-essential information.

**Strengths:**
The main strength lies in its innovative approach to filtering visual stimuli, leading to state-of-the-art performance in various benchmarks. Its method of using a task-conditioned codebook for selective attention mirrors human cognitive processes, enhancing task-specific learning. The paper also excels in demonstrating the method's generalizability and efficiency in different simulation environments. Additionally, the comprehensive analysis through ablation studies and qualitative assessments provides clear insights into the effectiveness of the proposed approach.

**Weaknesses:**
Despite its innovative approach, the paper's methodology is primarily applied to the EmbCLIP model, limiting its tested applicability across other general-purpose visual encoders. The absence of comparative baselines with other bottleneck methods like self-attention modules or auto-encoders is a notable gap. Additionally, the paper does not explore the model's potential in zero-shot or few-shot learning scenarios, which could be crucial for novel task adaptation. Lastly, the integration of human-like priors into the model, while mentioned, is not fully explored or implemented.

**Justification For Why Not Higher Score:**

I choose spotlight, but would not object if it is upgraded to an oral presentation. I chose spotlight because the method highly rely on embCLIP, could limit its generality. The paper tackles important problem and the approach is quite novel.

**Justification For Why Not Lower Score:**

The authors addressed the reviewers' concerns about their paper's limitations. They expanded their methodology to include the DINOv2 visual encoder, demonstrating the adaptability of their codebook module across different encoders. Additionally, they explored full-scale fine-tuning of the visual encoder and compared it with their codebook approach, showing the latter's superior performance. To investigate the significance of the codebook, they compared it with other bottleneck architectures like auto-encoders and self-attention modules, reaffirming the codebook's effectiveness.

---

### Decision · Program_Chairs · 2024-01-16

Accept (spotlight)